
# Ensemble Variational Assimilation as a Probabilistic Estimator. Part II: The fully non-linear case

Mohamed Jardak[1,2] and Olivier Talagrand[1]

[1]LMD/IPSL, CNRS, ENS, PSL Research University, 75231, Paris, France
[2]Data Assimilation and Ensembles Research & Development Group , Met Office, Exeter, Devon, UK

*Correspondence to:* M. Jardak (mohamed.jardak@metoffice.gov.uk)

**Abstract.** The method of Ensemble Variational Assimilation (EnsVAR) is implemented in fully nonlinear conditions on the Lorenz-96 chaotic 40-parameter model. In the case of strong-constraint assimilation, it requires to be used in association with the method of *Quasi-Static Variational Assimilation* (*QSVA*). It then produces ensembles which possess as much reliability and resolution as in the linear case, and its performance is at least as good as that of Ensemble Kalman Filter and Particle Filter. On the other hand, ensembles consisting of solutions that correspond to the absolute minimum of the objective function (as identified from the minimizations without QSVA) are significantly biased. In the case of weak-constraint assimilation, EnsVAR is fully successful without need to resort to QSVA.

## 1 Introduction

In the first Part of this work (Jardak and Talagrand (2017)), the technique of Ensemble Variational Assimilation (EnsVAR), which achieves exact Bayesian estimation in the conditions of linearity and Gaussianity, has been implemented on two chaotic toy models with small dimension. The first model was the 40-parameter model introduced by Lorenz (1996). A linearized version was used as reference for the case where exact Bayesianity is achieved. Experiments were then performed with the full nonlinear model over assimilation windows for which a linear approximation is almost valid for the temporal evolution of the uncertainty. Although non-linear effects are distinctly present, the statistical quality of the ensembles produced by EnsVAR is as good as in the linear case. The second model was the Kuramoto-Sivashinsky equation (Kuramuto and Tsuzuki (1975, 1976)). Similar conclusions were obtained.

EnsVAR is implemented in this second Part, still on the Lorenz (1996) model, over assimilation

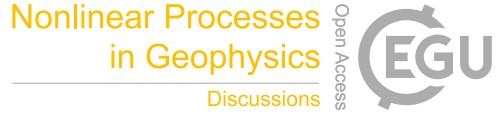



windows for which a linear approximation is no longer valid. It is implemented first in the strong-constraint case (Section 2), where it turns out to be necessary to use it together with the method of Quasi-Static Variational Assimilation (QSVA), introduced by Pires et al. (1996). The performance of

EnsVAR is compared with that of Ensemble Kalman Filter and Particle Filter in Section 3. EnsVAR is then implemented in the weak-constraint case (Section 4), where the use of QSVA turns out not to be necessary. Conclusions are drawn in Section 5. The general conclusion is that EnsVAR is as successful in nonlinear as in linear conditions.

Except when explicitly mentioned (and that will be the case mostly concerning the length of
the assimilation windows and the number $Nwin$ of realizations over which diagnostics are performed), the experimental set-up will be the same as in Part I. In particular, the size of the ensembles $Nens = 30$ will always be the same. And, unless specified otherwise, the space-time distribution of observations will also be the same (one complete set of observations of the state variable twice a day, with white-in-space-and-time noise, with standard deviation $\sigma = 0.63$).

Notations such as Eq. I.3 or Figure I.2 will refer to Equations or Figures of Part I.

## 2   Strong-constraint assimilation

Figure 1 shows the same diagnostics as Figures I-4 and I-5, for an experiment in which the length of the assimilation window is 10 days instead of 5. Comparison with Figures I-4 and I-5 shows an obvious degradation of the quality of the assimilation. The top panels (to be compared with the top
panels of Figure I-4) show that the dispersion of the minimizing solutions is now much larger. That dispersion is statistically much too large can be seen from the rank histogram (middle left panel), which has a distinct 'hump-backed' shape, meaning that the verifying truth is much too often located in the central part of the ensembles. The error from the truth is now larger than the observational error (bottom right panel), which is an obvious proof of failure of the assimilation process. The
reliability diagram (middle right panel) differs slightly, but markedly, from the diagram in Figure I-5 through its sigmoid shape. That can easily be verified to be consistent with the overdispersion seen on the rank histogram. And both Brier scores (bottom left panel) are significantly larger than in Figure I-5.

Another diagnostic is given in Figure 2, which shows the histogram of the minimizing values of
the objective function I-9 (the format is the same as in Figure I-3). The histogram is clearly bimodal. The values in the left mode have expectation 387.1 and standard deviation 18.8, in good agreement with the values of 400 and 20 indicated by the '$\chi^2$' linear theory (Eqs I-10-11, note that because of the increase of the length of the assimilation window from 5 to 10 days, the value of the parameter $p/2$ is now 400). This is to be noted since there is *a priori* no reason to expect that minimizations
that lead to the left mode correspond to errors $\epsilon_k$ and $\delta_k$ (Eqs I-7-8) distributed in such a way as to verify conditions (I-10-11). The right mode in Figure 2 is outside the linear approximation. It is also

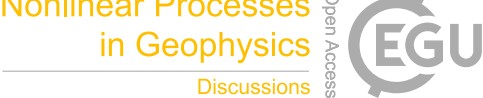



worth mentioning that out of 270,000 values of $\mathcal{J}_{min}$, only 96,330 were situated in the left mode.

These results tend to confirm the interpretation that was given of results obtained in Part I (see Figure I-7 and associated comments). This agrees with the the discussion and conclusions of the paper by Pires et al. (1996). Because of the chaotic character of the motion, the uncertainty on the position of the observed system is located on a folded subset in state space. The longer the observation period is, the more folded the uncertainty subset is. Secondary minima of the objective function I-9 may occur on the various folds (for more on this point, see figures 4 and 5 of Pires et al. (1996), and the discussion therein). With this interpretation, the left mode on Figure 2 corresponds to absolute minima, the right mode to secondary ones. Also, because of the longer assimilation window, the basin of attraction of the absolute minimum is narrower than that of Part I, and more minimizations lead to a secondary minimum.

Pires et al. (1996) showed that, even in the case of noisy observations of chaotic motion, the location of the absolute minimum of the objective function is not significantly affected by the observational noise. It makes therefore sense to locate that absolute minimum. To that end, they proposed the method of *Quasi-Static Variational Assimilation* (*QSVA*). In that method, the length of the assimilation window is gradually increased, keeping the same initial time, each new minimization being started from the result of the previous one.

This is what has been done here. Figure 3, which is in the same format as Figure 1, depicts the results produced by the use Quasi-Static Variational Assimilation over an overall 10-day assimilation window, with an increment of 1 day for the gradual increase of the assimilation window. The improvement over Figure 1 is obvious. The spread of the minimizing solutions (top two panels) is much smaller, the rank histogram (middle left) is almost perfectly flat, the reliability diagram (middle right) is much closer to the diagonal and both Brier scores (bottom left) have decreased. The estimation error (bottom right) is now, as it must, well below the observational error. The error in the ensemble mean (red curve) is now 0.1 at the middle point of the assimilation window, against 0.2 in Figure I-4, relative to assimilations over 5-day windows (without QSVA). This improvement must be due to the fact that more observations have been used. The rank histogram is also flatter than in the corresponding Figure I-5. That must be due mostly to the larger validating sample.

Figure 4, which is again in the same format as Figures 1 and 3, is relative to an 18-day QSVA (with still an increment of 1 day between successive assimilation windows). It confirms the previous conclusions. The estimation error, as well as both components of the Brier score, are reduced even further.

All these results show that Ensemble Variational Assimilation is successful, if implemented with QSVA, over long assimilation windows for which the tangent linear approximation is expected to fail. EnsVAR produces ensemble whith a high degree of statistical reliability. In addition, the accuracy of the estimated ensembles, as measured by resolution or by the error in the ensemble mean, is improved when the amount of information contained in the observations is increased.



It can be noted that, if EnsVAR is successful in 'non-linear' situations, it is not because of in-
trinsically non-linear character. Minimization of an objective function of form I-3 is a priori valid
for statistical estimation only in a linear situation. The success of EnsVAR probably results from
the fact that, through QSVA, it is capable of maintaining the current estimate of the flow within the
ever-narrower region of state space in which the tangent linear approximation is valid. If the tem-
poral density of the observations became so small, or alternatively if the dynamics of the observed
system became so non-linear that it would not be possible to 'jump' from one set of observations to
the next one within a linear approximation, EnsVAR would probably fail. This point will deserve
further study.

There is actually no reason to expect any strict link between the validity of the tangent linear
approximation and the possible statistical reliability of minimizing solutions that lie within that ap-
proximation (not to speak of their bayesianity). As already said, one can expect the *a posteriori*
bayesian probability distribution to be concentrated for long assimilation windows on a folded 'non-
linear' subset in state space. The bimodality of the histogram in Figure 2 has been interpreted as
separating the minimizations that lead to the absolute minimum of the objective function (left mode)
from those that lead to a secondary minimum (right mode). This suggests, without resorting to
QSVA, to retain only those minimizations corresponding to the left mode of the histogram. This of
course requires, if one wants to obtain ensembles with dimension $Nens$, a larger number of min-
imizations (even if there is of course no need to continue until convergence minimizations which
show at an early stage that they will lead to the right mode of the histogram).

This has been done on a set of $Nwin = 1443$ realizations (and over 10-day assimilation windows).
The results are shown on Figure 5, which presents the same diagnostics as those shown in the lower
four panels of Figure 1. The histogram (top left panel) shows a distinct bias towards low values of
the variable x, associated with a distinct underdispersion. This is confirmed by the RCRV, which has
positive mean 0.25 and standard deviation 1.29. It is also confirmed by the reliability diagram (top
right), which lies below the diagonal, and the error curves (bottom right), where the ratio between
the errors in the individual minimizations and in the mean of the ensembles is less than $\sqrt{2}$. The
former is qualitatively consistent with a bias towards low values of x, and the latter with both a bias
and an underdispersion. Finally, concerning the Brier score (bottom left), it is seen that the reliability
component is degraded with respect to QSVA (the resolution component, on the other hand, is not
significantly modified).

Clearly, this procedure is a failure as far as reliability is concerned. But it can also be noted that
the errors (bottom right panel) are smaller than those of Figure 3, especially at both extremities of
the assimilation window. The errors in the individual minimizations (blue curve) is 0.395 and 0.24
at the beginning and end of the assimilation window (against 0.47 and 0.28 respectively in Figure
3). As the errors in the ensemble means (red curves), they are 0.28 and 0.175 at the beginning and
end of the window (against 0.33 and 0.2 in Figure 3). There is less difference at the middle of the





window.

Judging from the above results, restricting the ensembles to minimizations that lead to the absolute minimum of the objective function degrades reliability, but improves to some extent the quadratic fit to reality. Now, the Bayesian expectation $\mathbb{E}(\mathbf{x}|\mathbf{z})$ is the deterministic function of the data vector $\mathbf{z}$

that minimizes the error variance on the state vector $\mathbf{x}$. Should the present results be confirmed, they would constitute an *a contrario* proof that QSVA, although it produces ensembles that possess high reliability, is not Bayesian.

### 3 Comparison with Ensemble Kalman Filter and Particle Filter

As in Part I, we compare the results produced by EnsVAR with those produced by Ensemble Kalman

Filter (EnKF) and Particle Filter (PF). Figures 6, 7 and 8 show the results of 10-day assimilations performed with, respectively, QSVA EnsVAR, EnKF and PF. The algorithms used for EnKF and PF are the same as in Part I. Except for the top left panel of Figure 6, the format of the figures is the same as the format of Figures 9-11 of Part I. The top left panel of 6, where the quantity on the horizontal axis is the spatial coordinate, shows the same diagnostics as the top left panels of Figures 3 and 4, but

for the final time of an assimilation window. It is again seen that the dispersion of the minimizing ensemble solutions is small. The top left panels of Figures 7 and 8 show also the dispersion of the minimizing solutions for one assimilation window, but as a function of time along the window. The dispersion is small for EnKF, but distinctly larger for PF. The other panels show on all three figures diagnostics at the end of the assimilations. Concerning the rank histogram (top right panels), it is

noisy for EnsVAR, but does not show otherwise any sign of disymmetry or inadequate spread. This is similar to what was observed in Part I at the end of 5 days of assimilation (Fig. I-9). On another hand, the histograms for EnKF and PF, in addition to being noisy, and again as in Part I (Figs I-10 and -11), have a distinct U-shape, which shows that the ensembles (although individually dispersed as shown by the top left panels) tend to 'miss their target'. Concerning the reliability diagrams (bottom

left panels), it is difficult to see visually any significant difference between the three algorithms. The Brier scores (bottom right panels) show similar performance for EnsVAR and EnKF, but distinctly poorer performance for PF. This again is similar to what was observed in Part I.

Similar results have been obtained, with the same conclusions, for longer assimilation periods (not shown).

EnsVAR shows therefore a slight advantage over EnKF, and a more distinct advantage over PF. This conclusion is however to be taken with some caution, and will be further discussed in the concluding section of the paper.



## 4 Weak-constraint assimilation

We present in this Section the results of experiments that have been performed in the 'weak-constraint'

case when the deterministic model I-6 is no longer considered as being exact. Following a standard

approach, we now assume that the truth is governed by the equation

$$\mathbf{x}_{k+1} = \mathfrak{M}(\mathbf{x}_k) + \mathbf{b}_k, \tag{1}$$

where $\mathbf{b}_k$ is a white-in-time stochastic noise with probability distribution $\mathcal{N}(0, \mathbf{Q}_k)$ at time $k$.

A typical experiment is as follows. A reference 'truth' $\mathbf{x}_k^r, k = 0, \cdots, K$ is created using (1) for a

particular realization of the noise $\mathbf{b}_k$. Noisy observations $\mathbf{y}_k$ are extracted from that reference truth

in the same way as in Part I (Eq. I-7). The data to be used in order to reconstruct the whole sequence

of states $\mathbf{x}_k^r$ now consist of the observations $\mathbf{y}_k$ and of the *a priori* estimates $\mathbf{w_k} = \mathbb{E}(\mathbf{b_k}) = \mathbf{0}$ of

the noise $\mathbf{b}_k$. The general expression I-3 for the objective function to be minimized then takes the

standard 'weak-constraint' form

$$\mathcal{J}(\boldsymbol{\xi}_0, \boldsymbol{\eta}_0, \boldsymbol{\eta}_1, \cdots, \boldsymbol{\eta}_{K-1}) = \frac{1}{2} \sum_{k=0}^{K} [H\boldsymbol{\xi}_k - \mathbf{y}_k]^T \mathbf{R}_k^{-1} [H\boldsymbol{\xi}_k - \mathbf{y}_k] + \frac{1}{2} \sum_{k=0}^{K-1} \boldsymbol{\eta}_k^T \mathbf{Q}_k^{-1} \boldsymbol{\eta}_k \tag{2}$$

subject to

$$\boldsymbol{\xi}_{k+1} = \mathfrak{M}(\boldsymbol{\xi}_k) + \boldsymbol{\eta}_k, k = 0, \cdots, K-1 \tag{3}$$

Implementation of ensemble variational assimilation as studied here requires to perturb both the

observations $\mathbf{y}_k$ and the estimates $\mathbf{w}_k$ according to their own error probability distribution. This

leads to minimize objective functions of the form

$$\mathcal{J}^{iens}(\boldsymbol{\xi}_0, \boldsymbol{\eta}_0, \boldsymbol{\eta}_1, \cdots, \boldsymbol{\eta}_{K-1}) = \frac{1}{2} \sum_{k=0}^{K} \left[ H\boldsymbol{\xi}_k - (\mathbf{y}_k^{iens})' \right]^T \mathbf{R}_k^{-1} \left[ H\boldsymbol{\xi}_k - (\mathbf{y}_k^{iens})' \right] + \frac{1}{2} \sum_{k=0}^{K-1} \left[ \boldsymbol{\eta}_k - (\mathbf{w}_k^{iens})' \right]^T \mathbf{Q}_k^{-1} \left[ \boldsymbol{\eta}_k - (\mathbf{w}_k^{iens})' \right]$$

$$\tag{4}$$

subject to condition (3)

In equation (4), $(\mathbf{y}_k^{iens})'$ is obtained, as in Eq. I-8, by perturbing the observation $\mathbf{y}_k$, while

$(\mathbf{w}_k^{iens})' \sim \mathcal{N}(0, \mathbf{Q}_k)$ is the perturbation around $\mathbf{w}_k = 0$.

As previously, for given reference solution $\mathbf{x}_k^r$ and observations $\mathbf{y}_k$, $Nens$ minimizations of ob-

jective functions of form (4) are performed with independent perturbations on $\mathbf{y}_k$ and $\mathbf{w}_k$. This is

repeated on $Nwin$ assimilation windows, with different $\mathbf{x}_k^r$ and observations $\mathbf{y}_k$. $Nens$ will always

be equal, as before, to 30. $Nwin$ will depend on the experiment.

The experiments have again been performed with the Lorenz 96 model (Eq. I-12). Experiments

performed with a linearized version of the model have produced results (not shown) that are entirely

consistent with the theory of the BLUE, within a numerical uncertainty which is similar to what has

been observed in Part I.

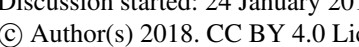



The covariance matrix $\mathbf{Q}_k$ of the stochastic noise has been taken equal to $q\mathbf{I}$, where $q$ is a positive scalar. Experiments have been performed to evaluate the impact of the value of $q$ on the predictability of the system. The value $q = 0.1$, which corresponds to a predictability time of about 10 days, is used in the sequel.

The experimental procedure is otherwise the same as before. In particular, the complete state vector is observed every 0.5 day, the observations being affected with uncorrelated unbiased Gaussian errors with the same variance $\sigma = 0.63$ as in the strong-constraint case.

The first conclusion that has been obtained is that QSVA is no longer necessary for achieving the minimization, at least up to assimilation windows of length 18 days (the largest value that has been tried). Clearly the presence of the additional noise penalty term in (4) has a regularizing effect which acts as a smoother of the objective function variations. This is in agreement with results already obtained by Fisher et al. (2005) in a study of weak-constraint variational assimilation.

Figure 9 shows the results obtained over an 18-day assimilation window (and, because of computational cost, only $Nwin = 1200$ realizations). The figure is to be compared with Figure 4, relative to 18-day strong-constraint variational assimilations. The top left panel, relative to one realization, shows the temporal evolution, over the assimilation window and at a particular gridpoint, of the truth and of the corresponding 30 minimizing solutions. It is seen that, if most of the latter closely follow the former, there are nevertheless a few 'outliers' ( two solutions out of 30 ). In view of what has already been said, these outliers must correspond to secondary minima of the objective function (thus showing departure from strict linearity). The rank histogram (top right) as well as the reliability diagram and the Brier scores (middle left and right respectively) show overall good performance, although not as good as in the strong-constraint case. In particular, both components of the Brier score are larger than their counterparts in Figure 4 (bottom left panel).

The bottom panels of Figure 9 show the RMS estimation error on the state variable $\mathbf{x}_k$ and on the model noise $\mathbf{b}_k$ (left and right panels respectively), as functions of time along the assimilation windows. In addition to the average RMS error in the individual minimizations (blue curves) and in the mean of the ensembles (red curves), the green curves (as in the bottom right panel of Figure 5) are in the ratio $1/\sqrt{2}$ to the blue curves. The error is generally smaller than the standard deviation of the corresponding 'observation error' (0.63 and 0.32 respectively). But it is actually larger in the individual minimizations at both ends of the assimilation window for the variable $\mathbf{x}_k$, and at the initial time of the window for the model noise $\mathbf{b}_k$. The coincidence of the red and green curves indicates statistical reliability. All curves in both panels show oscillations with a half-a-day period, with minima at observation times for the $\mathbf{x}_k$ error, and maxima for the $\mathbf{b}_k$ error. These oscillations are not visible in the individual minimizing solutions, nor in the mean of the ensembles (not shown), and become visible only on averages made over a large number of realizations. The origin of these oscillations can easily be understood. At observation times, the minimizing fields tend to fit closely the observations. In between observation times on the contrary, the minimization adjusts the model





noise so as to fit more closely the deterministic equation, with the consequence that the minimizing
fields drift from the truth. These oscillations show up only because the temporal distribution of the
observations is the same in all realizations of the assimilation. This interpretation is confirmed by
Figure 10, which shows the same diagnostics (without green curves) for weak-constraint assimila-
tions over 5-day windows, with observations once every day. The top and bottom panels show errors

in $\mathbf{x}_k$ and $\mathbf{b}_k$ respectively. The error in $\mathbf{x}_k$ is minimum at observation times, at which the error in $\mathbf{b}_k$
is maximum.

Figure 11, which is in the same format as Figure I-7, shows the distribution of (half) the minima
$\mathcal{J}_{min}$ of the objective functions. It is seen that, in superposition to a background of small minima,
a number of very large values are present. These are interpreted as corresponding (as in Figure

I-7) to secondary minima of the objective function (associated for instance with the outliers of the
top left panel of Figure 9). As concerns the theoretical '$\chi^2$' values, the number of observations of
the variable $\mathbf{x}_k$ has now increased to 37 x 40 = 1480. The use of weak constraint, which adds as
many parameters to be determined as parameters to be adjusted in the objective function (2), does
not modify the difference $p = m - n$. This leads for the expectation and standard deviation of $\mathcal{J}_{min}$

to the values $p/2 = 720$ and $\sqrt{p/2} = 26.8$. The sample values in the background of small minima
in Figure 11 are respectively 727.45 and 25.51, in good agreement with the theoretical values, and
with the interpretation that those small minima correspond to minimizing solutions that lie within
the range of the tangent linear approximation.

These results show that, although there are clearly imperfections (minimizations occasionally lead

to secondary minima), ensemble variational assimilation is on the whole very successful for weak-
constraint assimilation.

Figure 12 shows the compared performance of EnsVAR, EnKF and PF, evaluated over the last 13
days of the 18-day assimilation windows (this in order to eliminate the effects of the intialization of
EnKF and PF). The experimental conditions for EnKF and PF are exactly the same as for EnsVAR.

The three columns correspond, from left to right, to EnsVAR, EnKF and PF respectively. The rows
show, from top to bottom, the rank histograms, the reliability diagrams, and the two components of
the Brier score. The general performance of the three algorithms is similar. The only significant
difference is seen on the rank histograms. The histogram for EnsVAR is much flatter than the other
two histograms, which shows a distinct underdispersion of the ensembles. This is confirmed by the

standard deviations of the RCRV diagnostic, which are equal to 1.02, 1.14 and 1.11 for EnsVAR,
EnKF and PF respectively.

## 5  Discussion and conclusions

The principle of Ensemble Variational Assimilation (EnsVAR), which has been discussed in the
two Parts of this work, is very simple : perturb the data according to their own error probability



distribution and, for each set of perturbed data, perform a standard variational assimilation. In the linear and additive Gaussian case, this produces a sample of independent realizations of the (Gaussian) Bayesian probability distribution for the state of the observed system, conditioned by the data.

The primary purpose of this work was to study EnsVAR as a probabilistic estimator in conditions
(non-linearity and/or non-Gaussianity) where it cannot be expected to be an exact Bayesian estimator. Since the degree to which Bayesianity is achieved cannot be objectively evaluated, the weaker property of reliability has been evaluated instead. Standard scores, commonly used for evaluation of probabilistic prediction (rank histograms, reliability diagrams and associated Brier score, and in addition the Reduced Centred Random Variable) have been used to that end. The additional property
of resolution, *i. e.* the degree to which the estimation system is capable of *a priori* distinguishing between different outcomes, has also been evaluated (resolution component of the Brier score, root-mean-square error in the mean of the ensembles). Indeed, a secondary purpose of this work was to stress the importance, in the authors' minds, of evaluating ensemble assimilation systems as probabilistic estimators, particularly through the degree to which they achieve reliability and resolution.

The results presented in both parts of this paper show that EnsVAR is fundamentally successful in that, even in conditions where Bayesianity cannot be expected, it produces ensembles which possess a high degree of statistical reliability. Actually, the numerical scores for reliability that have been used are often as good, if not better, in situations where Bayesianity cannot be expected to hold than in situations where it holds. Better scores can be explained in the present situation only by better
numerical conditioning. The resolution, as measured by the RMS error in the mean of the ensembles, or by the resolution component of the standard Brier score, is also high (close to 1) .

In nonlinear strong-constraint cases, EnsVAR has been successful here only through the use of Quasi-Static Variational Assimilation, which significantly increases its numerical cost. However, in the weak-constraint case, QSVA has not been necessary, providing new evidence as to the favourable
effect, on numerical efficiency of assimilation, of introducing a weak constraint. At the same time, the comparison of the results shown in the right bottom panels of Figures 3 and 5, shows that EnsVAR, even when it has as high a degree of reliability as in purely linear and Gaussian situations, is not Bayesian.

Comparison with two other standard ensemble assimilation filters, namely Ensemble Kalman
Filter and Particle Filter, made at constant ensemble size, shows a superior or equal performance for EnsVAR, at least as concerns the dispersion of the ensembles.

If a code for variational assimilation is available, EnsVAR is very easy (if costly) to implement. It possesses the advantages and disadvantages of standard variational assimilation. Advantages are easy propagation of information both forward and backward in time (smoothing) and easy introduc-
tion of observations of new types and of temporal correlations between data errors. What is usually considered to be a major disadvantage of variational assimilation is the need for developing and





maintaining an adjoint code. Concerning that point, it must however be stressed that algorithms are being developed which might avoid the need for adjoints while keeping most of the advantages of variational assimilation.

EnsVAR, as it has been implemented here, is very costly in that it requires a very large number of iterative minimizations. The comparison with EnKF and PF, which has been made here at constant ensemble size, might have led to different conclusions if it had been made at, *e. g.*, constant computing cost. In addition, the particular versions of EnKF and PF that have been used here may not be, among the many versions that exist for both algorithms, the most efficient ones for the problem con-

sidered here. On the other hand, many possibilities exist for reducing the cost of EnsVAR, through simple parallelization or through use of the results of the first minimizations to speed up the following ones. The rapid development of numerical algorithmics makes it difficult to draw definitive conclusions at this stage as to the compared cost of various methods for ensemble assimilation.

EnsVAR, at it has been presented here, is almost uniquely defined on the basis of its principle. It

has been necessary to introduce only one arbitrary parameter for the experiments that have been described, namely the temporal increment (1 day) between successive assimilation windows in QSVA. Everything else is unambiguously defined once the principle of EnsVAR has been stated. This may of course not remain true in the future, but is certainly a distinct advantage to start with. On the other hand, EnsVAR is largely empirical, with the consequence that, should difficulties arise, conceptual

guidelines may be missing to solve these difficulties. The only thing that can be said at this stage is that EnsVAR is successful in nonlinear situations probably because it keeps the estimation problem within the basin of attraction of the absolute minimum of the objective function to be minimized.

One can also remark that EnsVAR, in the form in which it has been implemented here, and contrary to EnKF and PF, produces an ensemble of totally independent realizations of a same probability

distribution. It is difficult to say if that can be considered as a distinct advantage, but it is certainly not a disadvantage.

The problem of cycling EnsVAR for one assimilation window to the next one has not been considered here. The questions that arise in that respect range from the simplest one (is cycling necessary at all, or can one simply proceed by implementing EnsVAR over successive, possibly overlapping,

windows ?) to the question of carrying a 'background' ensemble from one window to the next, together with an associated error covariance matrix. In the latter case, the difficulties associated with localization and inflation, which have significantly complicated the development of EnKF, might arise again. One interesting possibility is to use the ideas of Assimilation in the Unstable Subspace (AUS), advocated by Trevisan and colleagues (see, Trevisan et al. (2010) and Palatella et al. (2013)),

in which the control variable of the assimilation is restricted to the modes of the system that have been unstable in the recent past, where the uncertainty on the state of the system is most likely to be concentrated. This approach, which mostly suited for toy models, is actively studied at present (Carrassi, Bocquet, *pers. com.*).





EnsVAR has been implemented here on a small dimension system. It has to be implemented on
larger dimension, physically more realistic, models. It has also to be compared with other ensemble assimilation methods, in terms of both intrinsic quality of the results and of cost efficiency. In addition to the many variants of Ensemble Kalman Filter and Particle Filter, one can mention the Metropolis-Hastings algorithm, which, as already said in the Introduction of Part I, possesses itself many variants. It has been used to many applications, most of which, if not all, are however relative to problems with small dimensions. It would be extremely interesting to study in performance in problems of assimilation for geophysical fluids. More recently, and in the continuation of Bardsley (2012), Bardsley et al. (2014) have proposed what they call the *Random-Then-Optimize* (*ROT*) algorithm. This defines a theoretical improvement on EnsVAR, based on an appropriate use of the Jacobian of the data operator. Systematic comparison of the performances of the many algorithms that now exist for ensemble assimilation, in particular in terms of their capability of achieving Bayesian estimation, will certainly be very instructive.

## 6   Acknowledgments

This work has been supported by Agence Nationale de la Recherche, France, through the Prevassemble and Geo-Fluids projects, as well as by the programme Les enveloppes fluides et l'environnement of Institut national des sciences de l'Univers, Centre national de la recherche scientifique, Paris. The authors acknowledge fruitful discussions with M. Bocquet and J. Brajard.





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


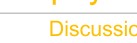
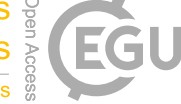


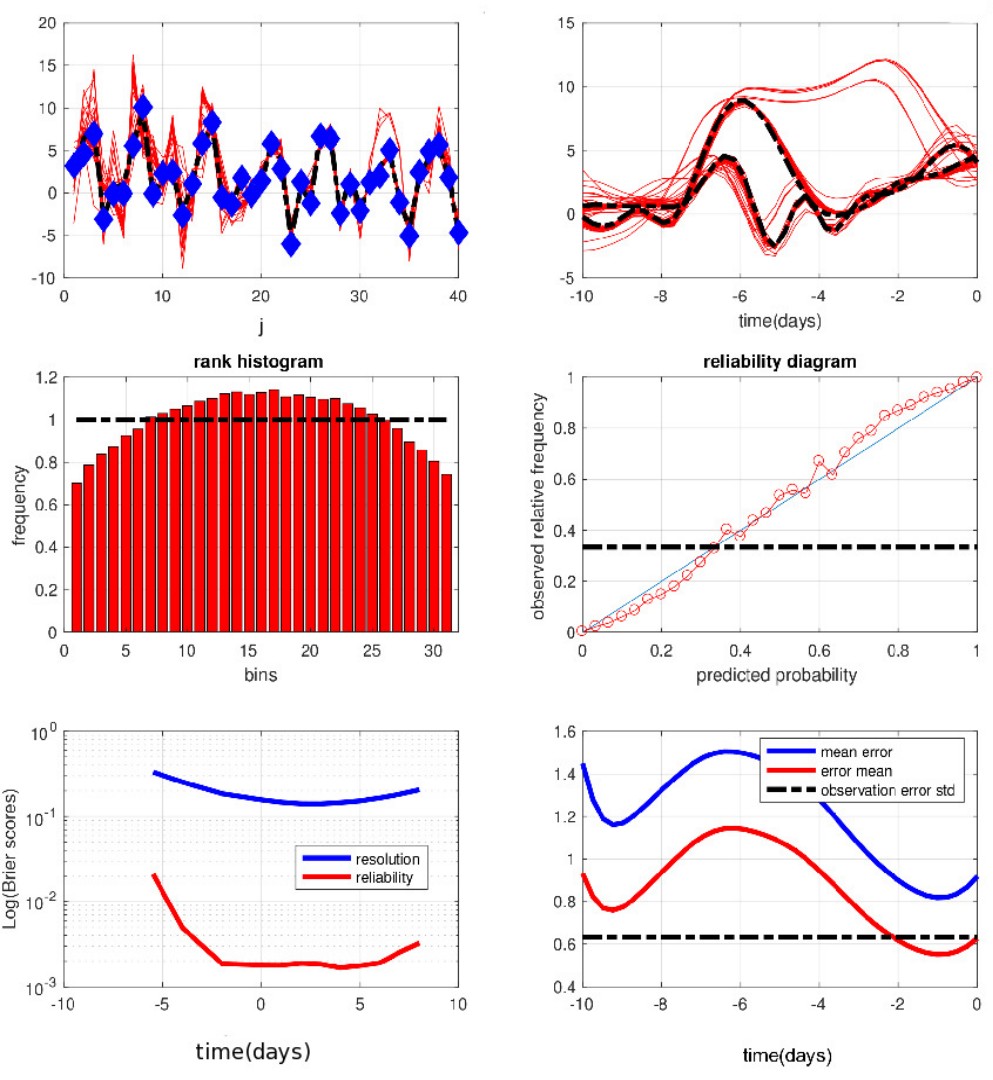

Fig. 1: Diagnostics of an experiment performed in the same conditions as for Figures I-4 and I-5, with the only difference that the length of the assimilation windows is now 10 days. Top left. Truth (dashed black curve), observations (blue dots) and minimizing solutions (red curves) at the initial time of one assimilation window, as functions of the spatial coordinate. Top right. Truth (dashed curves) and minimizing solutions (red curves) as functions of time at two gridpoints over one assimilation window. Middle left. Rank histogram for the variable x over all gridpoints and ensemble assimilations. Middle right. Reliability diagram for the event $\{\mathbf{x} < 1.14\}$, which occurs with frequency 0.35 (dashed-dotted horizontal curve), built over all gridpoints and ensemble assimilations. Bottom left. Components of the Brier score for the events $\mathscr{E} = \{\mathbf{x} < \tau\}$, evaluated over all gridpoints and ensemble assimilations, as functions of the threshold $\tau$ (red curve : reliability, blue curve : resolution). Bottom right. Average RMS errors to truth, as functions of time over assimilation windows. Blue curve : error in individual assimilations. Red curve : error in mean of ensembles (the black dash-dotted curve shows the standard deviation of observational error).





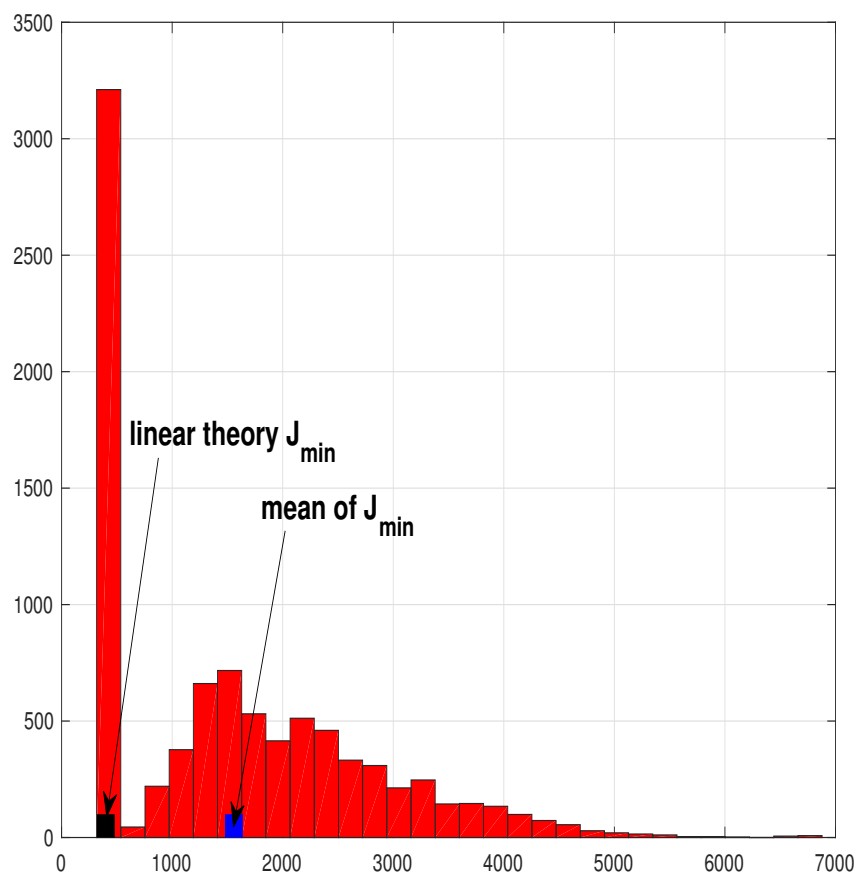

Fig. 2: Histogram of (half) the values of the minima of the objective function I-9, for the same experiment as in Figure 1.





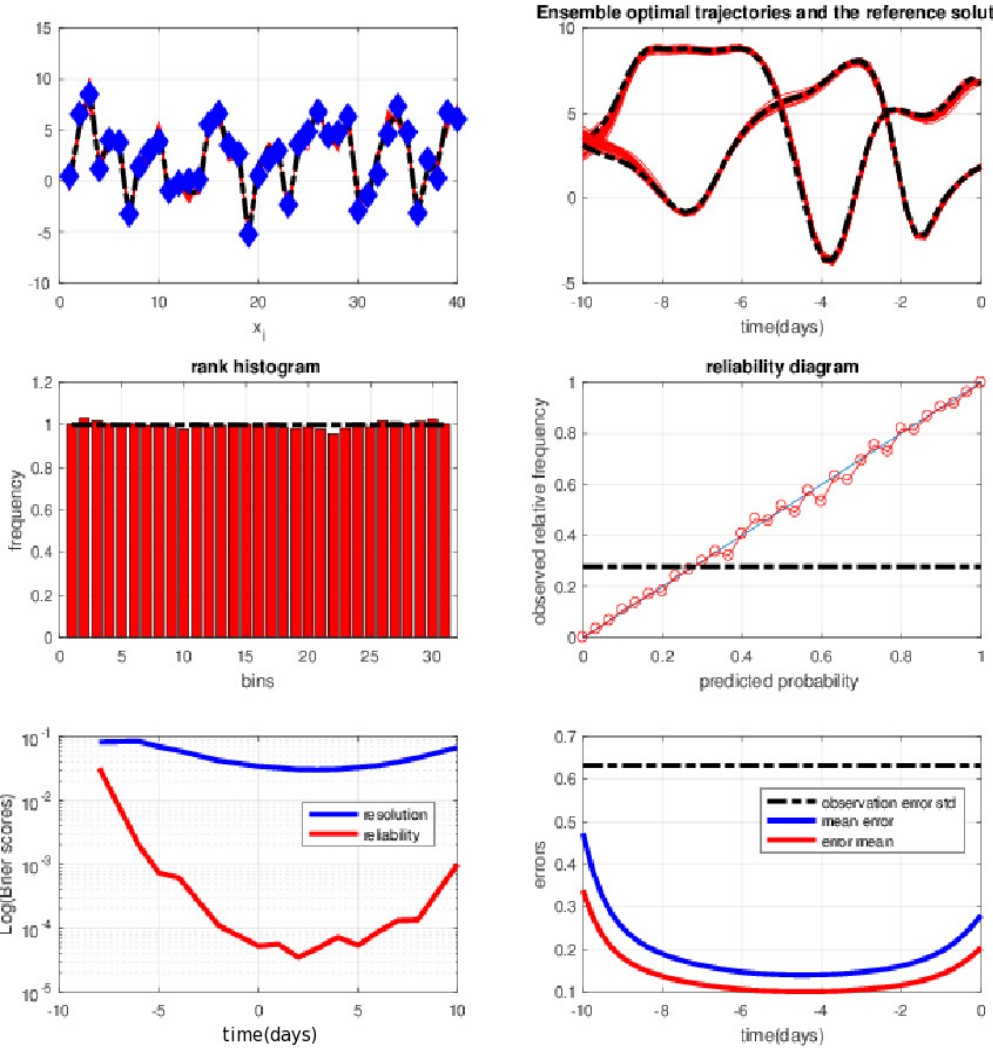

Fig. 3: Same as Figure 1, for Quasi-Static Variational Assimilations, with increase of the length of the successive assimilation windows by 1 day. All scores are computed for the ensembles obtained from the final minimizations performed over the whole 10-day assimilation windows. The reliability diagram is relative to the event $\mathscr{E} = \{\mathbf{x} < 0.94\}$, which occurs with frequency 0.29 (dashed-dotted horizontal curve).

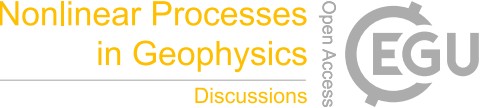



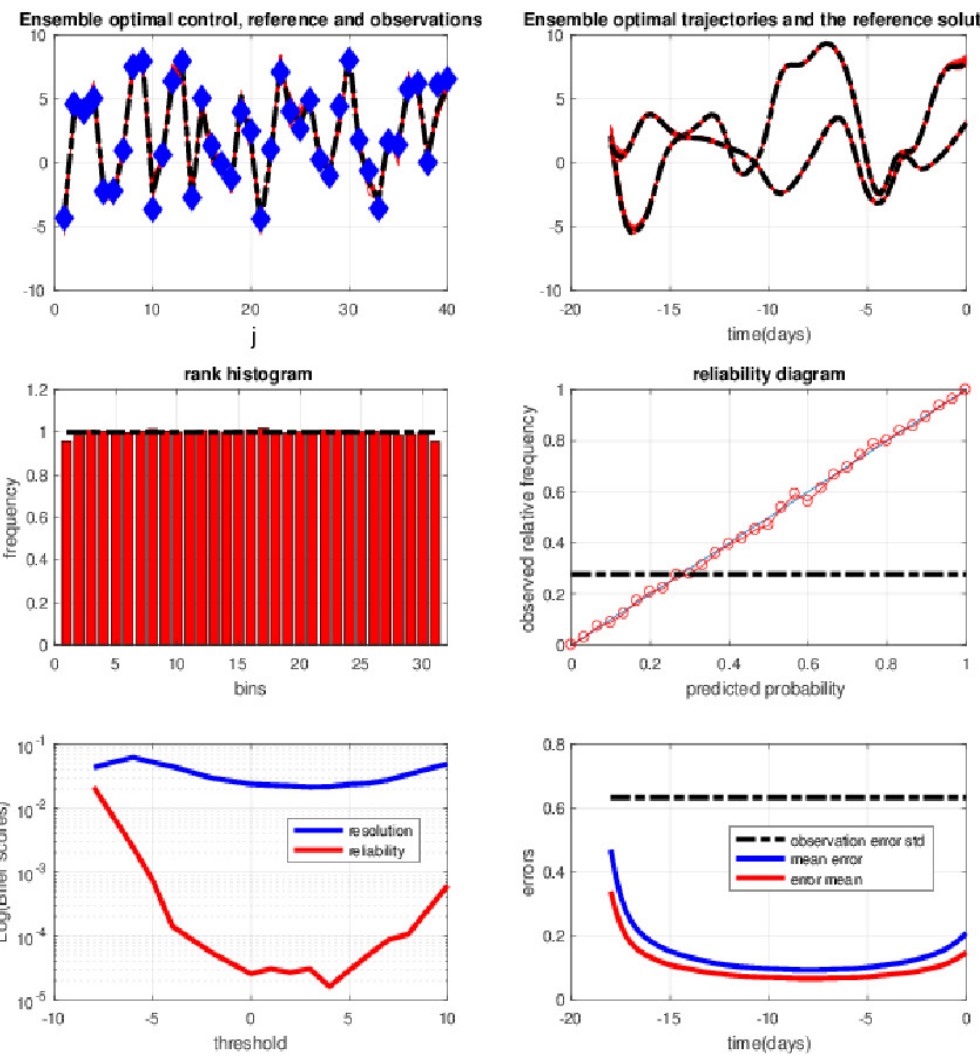

Fig. 4: Same as Figure 3, for Quasi-Static Variational Assimilations performed over 18-day assimilation windows, with increase of the length of the successive assimilation windows by 1 day. Reliability diagram for the event $\mathcal{E} = \{\mathbf{x} < 0.94\}$, which occurs with frequency 0.29 (dashed-dotted horizontal curve).





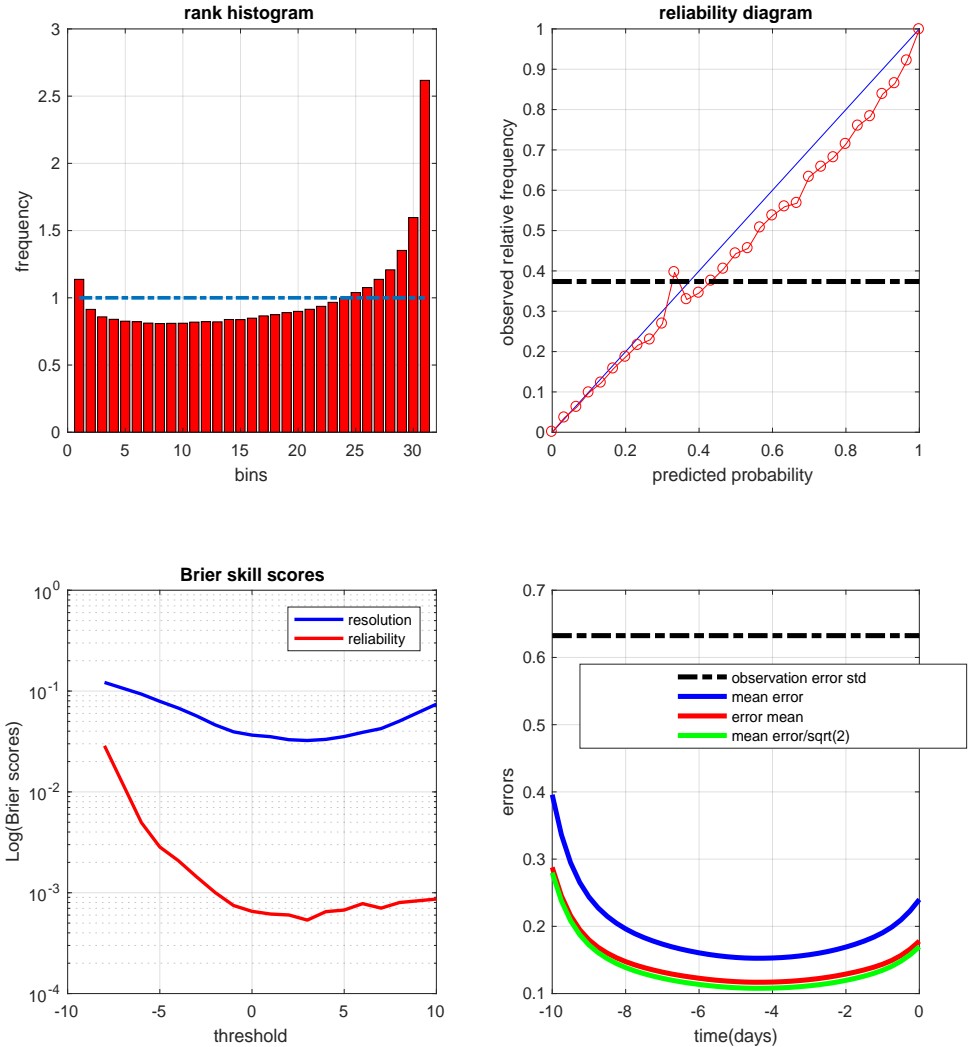

Fig. 5: Same as the lower four panels of Figure 1, restricted to minimizations that lead to the absolute minimum of the objective function I-9, as identified from the value of the minimum itself (see Figure 2). The diagnostics have been computed on 1443 realizations. The reliability diagram (top right) is relative to the event $\mathscr{E} = \{\mathbf{x} < 0.33\}$, which occurs with frequency 0.38. In the lower right panel, which shows the RMS estimation errors along the assimilation window, the green curve has been obtained by dividing the values on the blue curve by a factor $\sqrt{2}$.




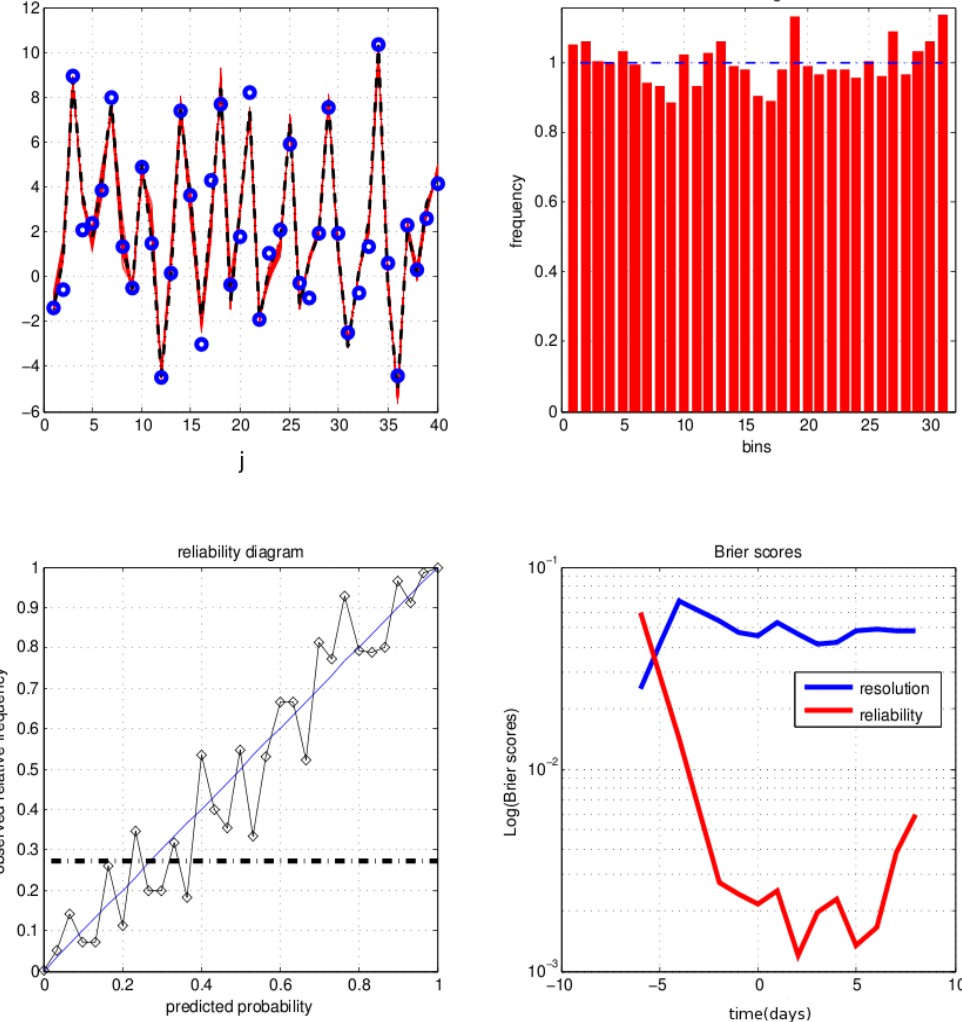

Fig. 6: Further diagnostics of 10-day ensemble assimilations performed with QSVA EnsVAR. All diagnotics are performed at the end of the assimilation windows. Top left. Truth (dashed black curve), observations (blue dots) and minimizing solutions (red curves), as functions of the spatial coordinate, at the end of one assimlation window. Top right. Rank histogram for the variable x over all gridpoints and ensemble assimilations. Bottom left. Reliability diagram for the event $\mathscr{E} = \{\mathbf{x} < 0.94\}$, which occurs with frequency 0.29 (dashed-dotted horizontal curve), built over all gridpoints and ensemble assimilations. Bottom right. Components of the Brier score (same format as in Figure 1).

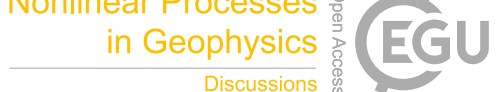



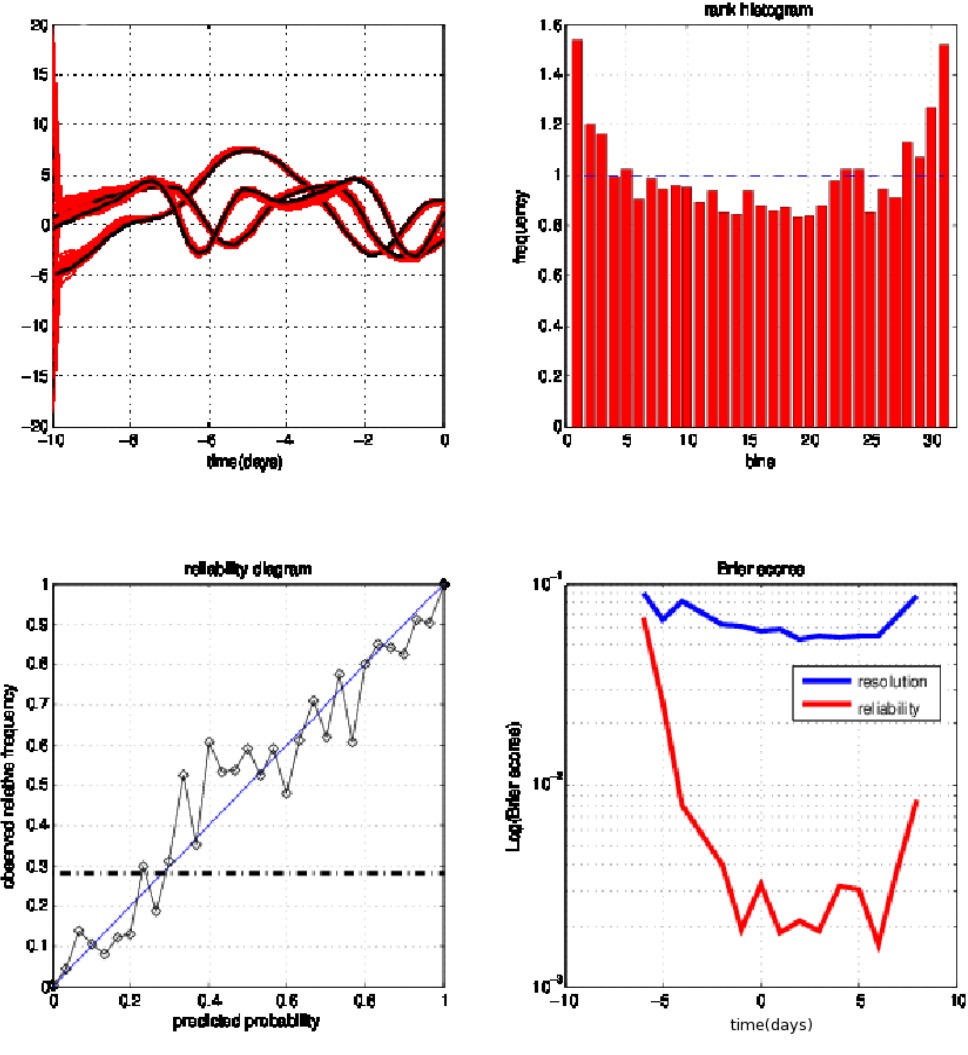

Fig. 7: Diagnostics for assimilations performed with EnKF. Top left. Temporal evolution, for one realization, of the truth at three gridpoints (black curves)and of the 30 corresponding ensemble solutions at the same points (red curves). The other three panels are in the same format as in Figure 6.




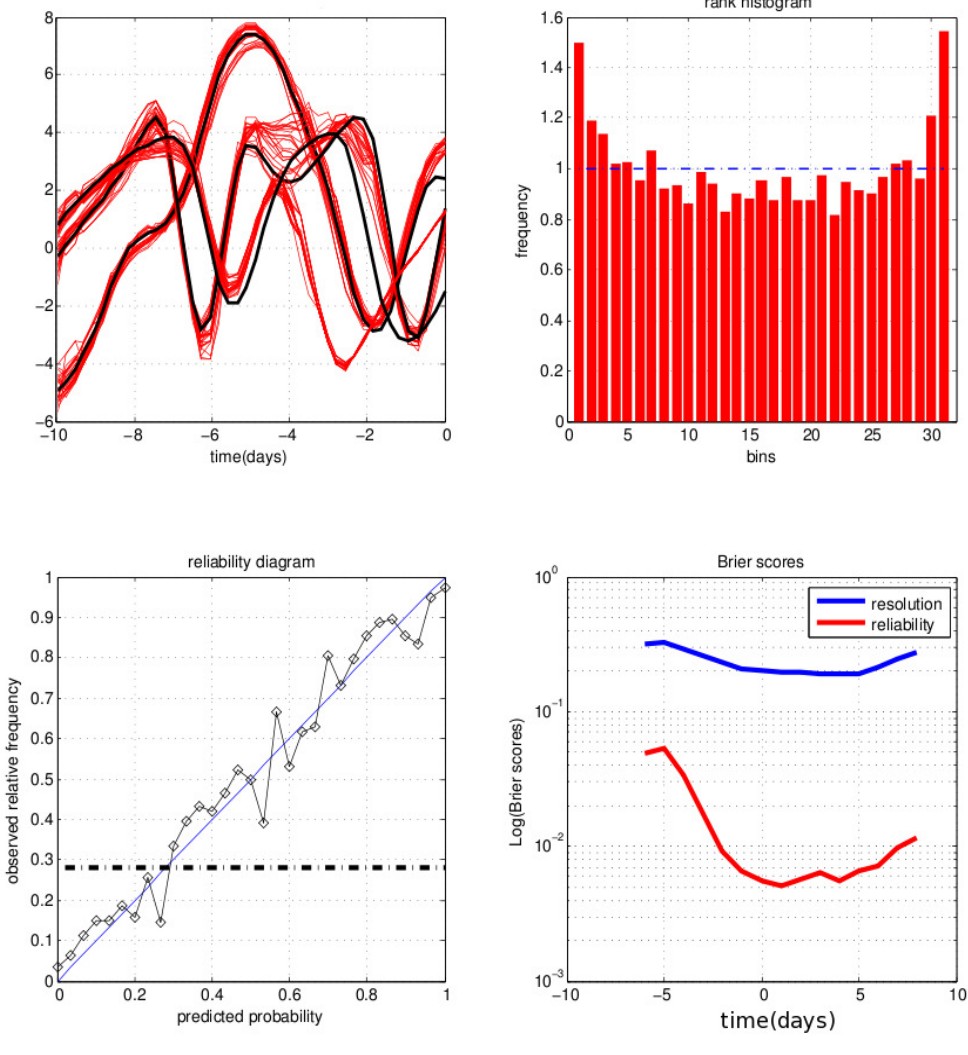

Fig. 8: Same as Figure 7, for assimilations performed with PF.



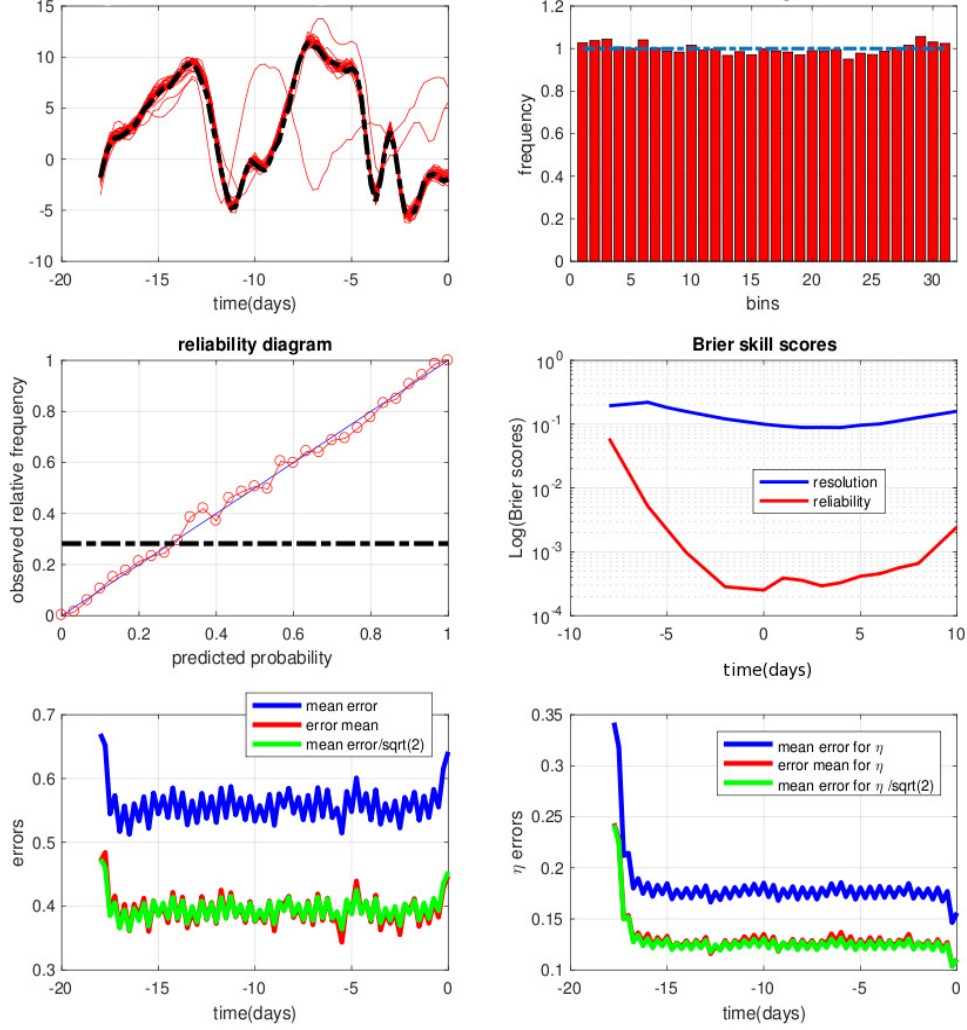

Fig. 9: Diagnostics of weak-constraint variational assimilations performed over 18-day assimilation windows. Top left. Truth (dashed curves) and minimizing solutions (red curves) as functions of time at one gridpoint over one assimilation window. Top right. Rank histogram for the variable $\mathbf{x}_k$ over all gridpoints and ensemble assimilations. Middle left. Reliability diagram for the event $\mathscr{E} = \{x > 1.02\}$, which occurs with frequency 0.27 (dashed-dotted horizontal curve), built over all gridpoints and ensemble assimilations. Middle right. Components of the Brier score (same format as in bottom left panel of Figure 1). Bottom panels. RMS estimation error on the state variable $\mathbf{x}_k$ and on the model noise $\mathbf{b}_k$ (left and right respectively), as functions of time along the assimilation windows. Blue curves : average RMS error in the individual minimizations. Red curves : average RMS error in the mean of the ensembles. The green curves are in the ratio $1/\sqrt{2}$ to the blue curves.





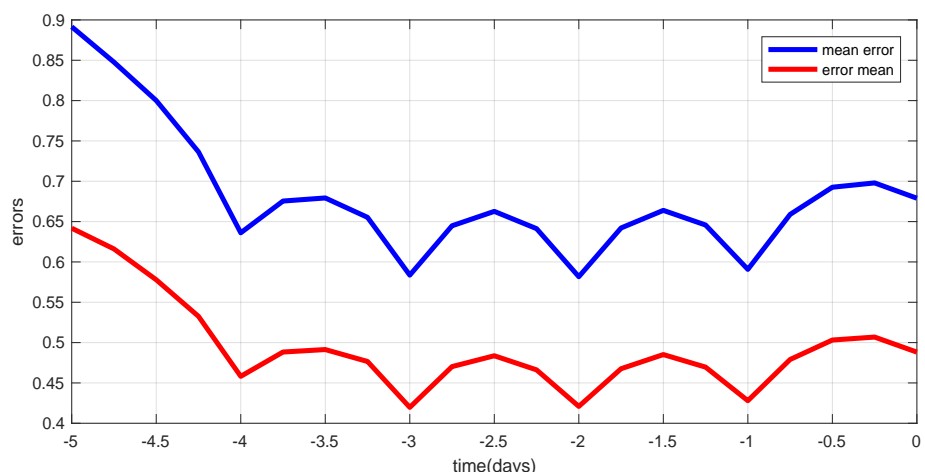

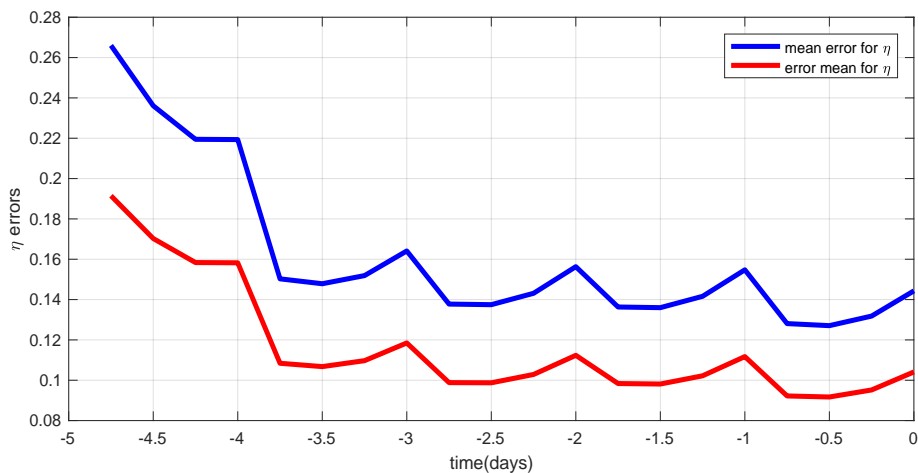

Fig. 10: RMS estimation errors on the state variable $\mathbf{x}_k$ and on the model noise $\mathbf{b}_k$ (top and bottom panels respectively) for weak-constraint variational assimilations performed over 5-day windows, with observations at times -5, -4, ..., 0.





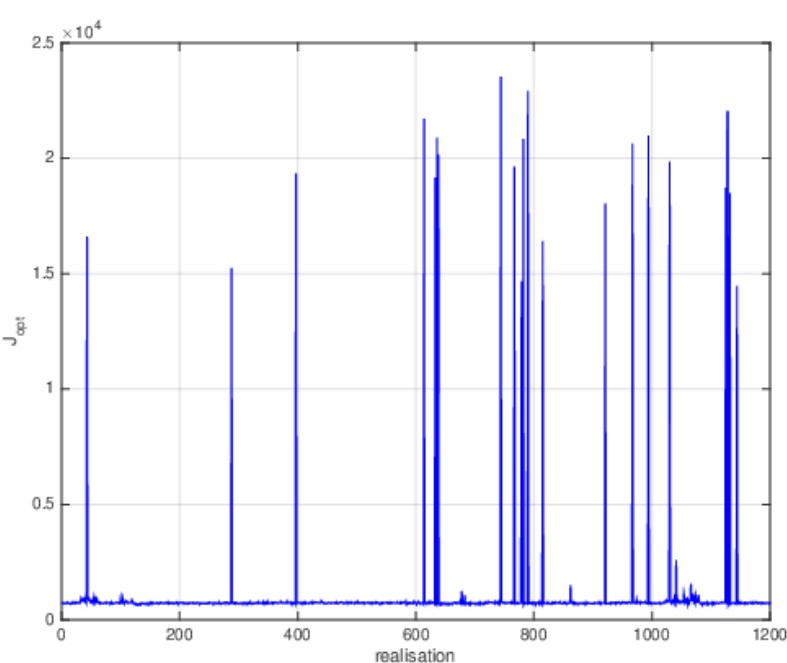

Fig. 11: Values of (half) the minima of the objective function for all realizations of the weak-constraint assimilations over 18-day windows.

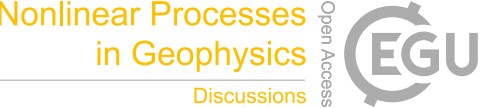



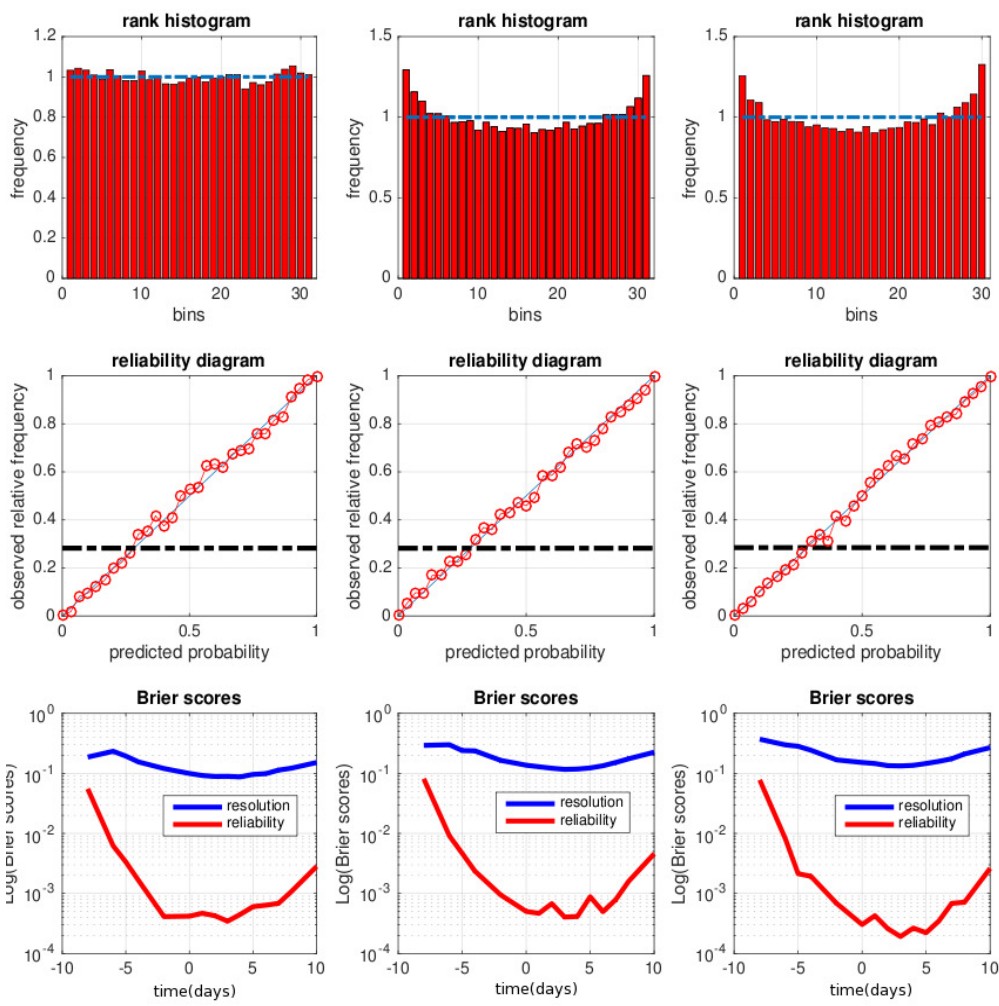

Fig. 12: Compared performance of EnsVAR, EnKF and PF (columns from left to right respectively) over the last 13 days of 18-day assimilation windows. Top row. Rank histogram. Middle row. Reliability diagram for the event $\mathscr{E} = \{x > 1.02\}$, which occurs with frequency 0.27. Bottom row. Reliability and resolution components of the Brier score for events $x < \tau$ as functions of the threshold $\tau$.