# Peer review of "Ensemble Variational Assimilation as a Probabilistic Estimator. Part II: The fully non-linear case"

_Nonlinear Processes in Geophysics, 2018_

## Referee Comment (RC1) · M. Bonavita (Referee) · 11 Feb 2018

The paper in subject is the second part of a study of the characteristics of the EnsVAR ensemble data assimilation as a probabilistic estimator. This second part aims to extend the results of the first part to the fully nonlinear case. While the basic methodology follows the one used in the first part of the study, the robustness of some the results presented in this second part appears more questionable and some issues need to be explored further, at least to the mind of this Reviewer, in order for the paper to be acceptable for publication. In the following I detail these concerns.

1) In Sect. 3 the Authors compare results of the QSVA EnsVAR, EnKF, PF. Not sur-

prisingly, QSVA EnsVAR shows better results as a probabilistic estimator and also for more standard resolution measures. This is unsurprising, to my mind, because this comparison is not fair. As the Authors noted, the costly QSVA extension is needed to keep EnsVAR assimilation in an approx. linear error evolution regime and thus guarantee good behaviour in this long-window assimilation set-up. To compare apples with apples the Authors should directly compare results of the standard EnsVAR algorithm at the end of the window with those of EnKF and PF. Additionally, it would also be of interest to compare results of QSVA EnsVAR with those of an EnKF whose assimilation is run on shorter assimilation windows, to guarantee linear behaviour, and then cycled;

2) In Sect. 4 on weak-constraint assimilation, I understand that the model error perturbations are drawn from the same error distribution whose covariance is used in the 4D-Var cost function. If this is correct, this is a significant limitation on the potential applicability of the results, as the difficulty in obtaining realistic characterizations of Q is probably the most important cause of the limited success of weak-constraint 4D-Var in realistic applications;

3) In the last paragraph of Sect. 4, the Authors explain that the performance of EnsVAR, EnKF, PF in the weak-constraint case appears in terms of reliability measures (e.g., rank histograms). This could depend on localization used in the EnKF, for example. Have the Authors explored this parameter space?

4) Lines 310-311: " ...many possibilities exist for the reducing the cost of EnsVAR, through simple parallelization or ...". I am puzzled on how parallelization can reduce the computational cost of EnsVAR. Maybe the Authors meant clock time?

5) Lines 319-320: "On the other hand, EnsVAR is largely empirical, with the consequence that, should difficulties arise, conceptual guidelines may be missing to solve these difficulties." I struggle to see what these difficulties might be. In the linear case, EnsVAR (aka EDA) is constructed so as to be a consistent statistical estimators assuming the input data errors are correctly sampled. In the nonlinear case, its behaviour will

depend on the amount of nonlinearity and the ability to track the true global solution. In this respect, EnsVAR is as empirical as the EnKF.

6) Lines 339-340: "EnsVAR has been implemented here on a small dimension system. It has to be implemented on larger dimension, physically more realistic models.". I suspect the Authors mean QSVA-EnsVAR in this context. Standard EnsVAR has been running at ECMWF and MeteoFrance for a number of years.

---

## Referee Comment (RC2) · M. Bocquet (Referee) · 12 Mar 2018

Please see my review of the companion manuscript (Jardak and Talagrand, 2018). I found the discussions of the numerical results of this part II paper equally well acute and clever. Section 4 on the weak-constraint EnsVAR is very nice, as well as Section 5 with a strong discussion and conclusion part.

Yet, most of the remarks that I made on the form of the companion paper still apply and need to be addressed.

Like for the first and companion paper, I believe that a minor revision of the manuscript is necessary to address a few flaws and a list of very minor points. In particular, a few references are missing.

Specific remarks, in connection, or not, to the previous remarks are:

1. Abstract, line 1: You should mention here that EnsVAR is equivalent to EDA. The main issue is the confusion that it may generate, and the fact that, because you fail to refer to EDA in the abstract, you will restrict your potential readership.

2. Abstract, lines 3-4: If you had cycled the analysis, you would have observed that QSVA is not as mandatory, expect maybe for very long windows. So I believe you should mitigate the statement.

3. Abstract, line 9: "without need to resort to QSVA" $\longrightarrow$ "without the need for QSVA".

4. line 19: "Kuramuto" $\longrightarrow$ "Kuramoto", as well as in both references by Yoshiki Kuramoto et al.

5. lines 24-25: "The performance of EnsVAR is compared with that of Ensemble Kalman Filter and Particle Filter in Section 3.": again, out of a specific context, this does not make much sense in the absence of cycling, proper tuning of the methods, and so on.

6. line 28: "successful in nonlinear as in linear conditions.": it always depends on how long the data assimilation window is. As any other method, EnsVAR is bound to fail for very large windows.

7. line 33: "twice a day": please mention that this corresponds to $0.10$ time units since the Lorenz model is primarily defined in those units.
8. line 35: Fine with the "I." but the notation for referring to equations is not consistent throughout the manuscript and does not follow the Nonlinear Processes in Geophysics guidelines.

9. line 59: "the the" $\longrightarrow$ "the".

10. Section 2: Nice results. Similar and consistent results have been obtained, which should be briefly mentioned. Bocquet and Sakov (2013) have obtained very similar results with the iterative ensemble Kalman smoother (IEnKS) with the same window of $10$ days, a time-interval of $1$ day (as opposed to twice a day), an ensemble of $20$ members and $\sigma = 1$: see Figure 4 of Bocquet and Sakov (2013). In particular the MDA IEnKS ($S = 1$), which is quasi-static, outperforms the SDA IEnKS which (in this reference) is not quasi-static. Other directly relevant references worth citing about quasi-static EnVar methods are Goodliff et al. (2015) and Carrassi et al. (2017).

11. lines 82-83: "This improvement must be due to the fact that more observations have been used.": above all this is due to the fact that the middle point is farther apart from the end of the window, so that fresher observations have a strong information content leveraged by the unstable modes of the dynamics. This has been shown in Bocquet and Sakov (2014).

12. line 105: "bayesianity" $\longrightarrow$ "Bayesianity".

13. line 106: "bayesian" $\longrightarrow$ "Bayesian".

14. lines 102-113: Part of this analysis coincides with that of H. Abarbanel and his collaborators. I believe you should at least refer to one of their paper, for instance Ye et al. (2015).

15. line 115: "shown on Figure 5" $\longrightarrow$ "shown in Figure 5".

16. line 129: "As the errors in the ensemble means...": I believe you mean "error standard deviations of the ensemble means".

17. lines 151-153: My experience is that, on the contrary, rank histograms of a deterministic EnKF (ETKF specifically) are ∩-shape (the ensemble is overdispersed). The difference might be due to the nature of the EnKF, the fact that your EnKF run is not long enough, or simply, that your inflation is insufficient. Moreover, once again, localisation is unnecessary with an ensemble of $30$ members and may be detrimental to the quality of the ensemble. Anyway, the ∪-shape that you have obtained for the EnKF is note a generality.

18. line 167: b is a notation usually reserved for a possible bias in VarBC.

19. line 175, Eq. (2): Assuming $H$ is linear, it should read $\mathbf{H}$.

20. line 182: Dot missing at the end of the sentence.

21. 195: "which corresponds to a predictability time of about 10 days": interesting. Can you please develop?

22. lines 252-255: How did you implement model noise in the EnKF and the PF? This should be described.

23. line 263: "simple :" ⟶ "simple:".

24. lines 277-279: Yes, that is the most important added value of this couple of papers and should be emphasised in the abstract of the first manuscript.

25. lines 294-296: In general, no claim can be made as to the accuracy of these methods (with the goal to estimate the truth) in the absence of cycling.

26. lines 305-310: I already know for a fact (Bocquet and Sakov, 2013, 2014) that proper cycling would very significantly reduce the number of iterations. This should be mentioned.

27. lines 328-330: "is cycling necessary at all, or can one simply proceed by implementing EnsVAR over successive, possibly overlapping, windows ?": This question has already a detailed answer in (Bocquet and Sakov, 2013, 2014) and subsequent references. To anticipate a question: yes, many of the conclusions obtained with the IEnKS would apply to EnsVAR. In essence: no, it is not absolutely necessary, but it would numerically help a lot to cycle the background (fewer iterations) and would yield a better accuracy.

28. line 338: "(Carrassi, Bocquet, pers. com.)": this has now been published (Bocquet and Carrassi, 2017).

29. line 347: "ROT" $\longrightarrow$ "RTO".

30. lines 348-349: "This defines a theoretical improvement on EnsVAR, based on an appropriate use of the Jacobian of the data operator." Liu et al. (2017) have already shown on a higher dimensional example that RTO might become inefficient (it is likely to be ultimately subject to the curse of dimensionality) as reported in their experiments and conclusions. This could be mentioned.

**References**

Bardsley, J.M., Solonen, A., Haario, H., Laine, M., 2014. Randomize-then-optimize: A method for sampling from posterior distributions in nonlinear inverse problems. SIAM J. Sci. Comput. 36, A1895–A1910.

Bocquet, M., Carrassi, A., 2017. Four-dimensional ensemble variational data assimilation and the unstable subspace. Tellus A 69, 1304504. doi:10.1080/16000870.2017.1304504.

Bocquet, M., Sakov, P., 2013. Joint state and parameter estimation with an iterative ensemble Kalman smoother. Nonlin. Processes Geophys. 20, 803–818. doi:10.5194/npg-20-803-2013.

Bocquet, M., Sakov, P., 2014. An iterative ensemble Kalman smoother. Q. J. R. Meteorol. Soc. 140, 1521–1535. doi:10.1002/qj.2236.

Carrassi, A., Bocquet, M., Hannart, A., Ghil, M., 2017. Estimating model evidence using data assimilation. Q. J. R. Meteorol. Soc. 143, 866–880. doi:`10.1002/qj.2972`.

Goodliff, M., Amezcua, J., van Leeuwen, P.J., 2015. Comparing hybrid data assimilation methods on the Lorenz 1963 model with increasing non-linearity. Tellus A , 26928doi:`10.3402/tellusa.v67.26928`.

Jardak, M., Talagrand, O., 2018. Ensemble variational assimilation as a probabilistic estimator. Part I: The linear and weak non-linear case. Nonlin. Processes Geophys. Discuss. 2018, 1–39. doi:`10.5194/npg-2018-5`.

Liu, Y., Haussaire, J.M., Bocquet, M., Roustan, Y., Saunier, O., Mathieu, A., 2017. Uncertainty quantification of pollutant source retrieval: comparison of bayesian methods with application to the Chernobyl and Fukushima-Daiichi accidental releases of radionuclides. Q. J. R. Meteorol. Soc. 143, 2886–2901. doi:`10.1002/qj.3138`.

Oliver, D.S., He, N., Reynolds, A.C., 1996. Conditioning permeability fields to pressure data, in: ECMOR V-5th European Conference on the Mathematics of Oil Recovery, pp. 259–269.

Ye, J., Rey, D., Kadakia, N., Eldridge, M., Morone, U.I., Rozdeba, P., Abarbanel, H.D.I., Quinn, J.C., 2015. Systematic variational method for statistical nonlinear state and parameter estimation. Phys. Rev. E 92, 052901. doi:`10.1103/PhysRevE.92.052901`.

---

## Author Comment (AC1) · 9 Apr 2018

We thank M. Bonavita for his comments and suggestions. We give below a first response to them.

The paper in subject is the second part of a study of the characteristics of the EnsVAR ensemble data assimilation as a probabilistic estimator. This second part aims to extend the results of the first part to the fully nonlinear case. While the basic methodology follows the one used in the first part of the study, the robustness of some the results presented in this second part appears more questionable and some issues need to be explored further, at least to the mind of this Reviewer, in order for the paper to be

acceptable for publication. In the following I detail these concerns.

1) In Sect. 3 the Authors compare results of the QSVA EnsVAR, EnKF, PF. Not surprisingly, QSVA EnsVAR shows better results as a probabilistic estimator and also for more standard resolution measures. This is unsurprising, to my mind, because this comparison is not fair. As the Authors noted, the costly QSVA extension is needed to keep EnsVAR assimilation in an approx. linear error evolution regime and thus guarantee good behaviour in this long-window assimilation set-up. To compare apples with apples the Authors should directly compare results of the standard EnsVAR algorithm at the end of the window with those of EnKF and PF. Additionally, it would also be of interest to compare results of QSVA EnsVAR with those of an EnKF whose assimilation is run on shorter assimilation windows, to guarantee linear behaviour, and then cycled.

We do not fully understand what the referee means. We consider our comparison is fair in the sense that we compare three algorithms that have used the same information (same model, same observations and same statistics as to the associated error; that is only what matters in the last resort). As for standard EnsVAR algorithm (i.e., without QSVA), it fails over a 10-day window, as shown by Fig. 1. And, from what we understand, the referee suggests to 'cycle' EnKF. To us, the latter, which is sequential, is cycled by construction.

2) In Sect. 4 on weak-constraint assimilation, I understand that the model error perturbations are drawn from the same error distribution whose covariance is used in the 4D-Var cost function. If this is correct, this is a significant limitation on the potential applicability of the results, as the difficulty in obtaining realistic characterizations of Q is probably the most important cause of the limited success of weak-constraint 4D-Var in realistic applications.

The referee raises a very important question, but it goes well beyond the scope of our present papers. An assimilation algorithm must first be evaluated in conditions where the errors affecting the data follow the same statistics (first- and second-order

moments) that are used in the assimilation. That is what we have done. Skipping that step would make the interpretation of results much more difficult. The same question arises concerning observation errors. We will stress that our twin experiments are fully 'consistent' concerning the errors on the data.

3) In the last paragraph of Sect. 4, the Authors explain that the performance of EnsVAR, EnKF, PF in the weak-constraint case appears in terms of reliability measures (e.g., rank histograms). This could depend on localization used in the EnKF, for example. Have the Authors explored this parameter space?

This remark is appropriate. But, no (except for basic elementary checks at the start), we have not done any comparison or tuning on the parameters of both EnKF and PF. This is actually mentioned ll. 308-310. A similar question has been raised by referee 2's comments on our first paper (see his comment 36 and our response).

4) Lines 310-311: " ...many possibilities exist for the reducing the cost of EnsVAR, through simple parallelization or ...". I am puzzled on how parallelization can reduce the computational cost of EnsVAR. Maybe the Authors meant clock time?

Yes, you're right. Thanks for the remark.

5) Lines 319-320: "On the other hand, EnsVAR is largely empirical, with the consequence that, should difficulties arise, conceptual guidelines may be missing to solve these difficulties." I struggle to see what these difficulties might be. In the linear case, EnsVAR (aka EDA) is constructed so as to be a consistent statistical estimators assuming the input data errors are correctly sampled. In the nonlinear case, its behaviour will depend on the amount of nonlinearity and the ability to track the true global solution. In this respect, EnsVAR is as empirical as the EnKF.

Yes, we do not know either at this stage what theses difficulties might be. But one must always be ready to encounter unexpected difficulties. And, yes, EnKF is also empirical, and the remark we make about EnsVAR also applies to EnKF.

6) Lines 339-340: "EnsVAR has been implemented here on a small dimension system. It has to be implemented on larger dimension, physically more realistic models.". I suspect the Authors mean QSVA-EnsVAR in this context. Standard EnsVAR has been running at ECMWF and MeteoFrance for a number of years.

No, we did not necessarily mean QSVA-EnsVAR (although that of course may be part of the work to be done). We meant systematic assessment of EnsVAR as a probabilistic estimator. We will rephrase our statement.

---

## Author Comment (AC2) · 9 Apr 2018

We thank M. Bocquet for his comments and suggestions. We give below a first response to some of these.

Like for the first and companion paper, I believe that a minor revision of the manuscript is necessary to address a few flaws and a list of very minor points. In particular, a few references are missing.

Specific remarks, in connection, or not, to the previous remarks are:

5.  lines 24-25: "The performance of EnsVAR is compared with that of Ensemble

Kalman Filter and Particle Filter in Section 3.": again, out of a specific context, this does not make much sense in the absence of cycling, proper tuning of the methods, and so on.

We have mentioned (ll. 305-310) that the comparison with EnKF and PF cannot certainly be considered as definitely conclusive. But it is certainly instructive, for instance in that it suggests that there are no major differences between the results produced by the three methods that have been compared. And we do not understand why the referee considers that 'this does not make much sense in the absence of cycling' (see our response to his specific remark 34 about paper 1). And 'proper tuning of the methods' could be an endless task.

6. line 28: "successful in nonlinear as in linear conditions.": it always depends on how long the data assimilation window is. As any other method, EnsVAR is bound to fail for very large windows.

We will qualify our statement by saying that it is valid for the time windows we have considered, but not necessarily for longer ones. But is not clear to us why any method is bound to fail for very large windows. Failure is certainly to be expected for strong constraint assimilation implemented with an erroneous model. But why should it be in the case of weak constraint ?

10. Section 2: Nice results. Similar and consistent results have been obtained, which should be briefly mentioned. Bocquet and Sakov (2013) have obtained very similar results with the iterative ensemble Kalman smoother (IEnKS) with the same window of 10 days, a time-interval of 1 day (as opposed to twice a day), an ensemble of 20 members and $\sigma = 1$: see Figure 4 of Bocquet and Sakov (2013). In particular the MDA IEnKS (S = 1), which is quasi-static, outperforms the SDA IEnKS which (in this reference) is not quasi-static. Other directly relevant references worth citing about quasi-static EnVar methods are Goodliff et al. (2015) and Carrassi et al. (2017).

Thanks for mentioning. We will look at those references.

11. lines 82-83: "This improvement must be due to the fact that more observations have been used.": above all this is due to the fact that the middle point is farther apart from the end of the window, so that fresher observations have a strong information content leveraged by the unstable modes of the dynamics. This has been shown in Bocquet and Sakov (2014).

Thanks also for mentioning.

14. lines 102-113: Part of this analysis coincides with that of H. Abarbanel and his collaborators. I believe you should at least refer to one of their paper, for instance Ye et al. (2015).

Thanks again.

25. lines 294-296: In general, no claim can be made as to the accuracy of these methods (with the goal to estimate the truth) in the absence of cycling ;

26. lines 305-310: I already know for a fact (Bocquet and Sakov, 2013, 2014) that proper cycling would very significantly reduce the number of iterations. This should be mentioned.

Is that last comment the basis for your insistence on cycling (in the comment just before and in comment 5 ?

27. lines 328-330: "is cycling necessary at all, or can one simply proceed by implementing EnsVAR over successive, possibly overlapping, windows ?": This question has already a detailed answer in (Bocquet and Sakov, 2013, 2014) and subsequent references. To anticipate a question: yes, many of the conclusions obtained with the IEnKS would apply to EnsVAR. In essence: no, it is not absolutely necessary, but it would numerically help a lot to cycle the background (fewer iterations) and would yield a better accuracy.

Thanks once more.

30. lines 348-349: "This defines a theoretical improvement on EnsVAR, based on an appropriate use of the Jacobian of the data operator." Liu et al. (2017) have already shown on a higher dimensional example that RTO might become inefficient (it is likely to be ultimately subject to the curse of dimensionality) as reported in their experiments and conclusions. This could be mentioned.

All right. Thanks.

---

## Author Response (AR1)

**Answers to referees: npg-2018-6, 2018**

Mohamed Jardak & Olivier Talagrand

**1    To referee 1:**

We thank M. Bonavita for his comments and suggestions. These are printed below in black, and our responses in red.

1. In Sect. 3 the Authors compare results of the QSVA EnsVAR, EnKF, PF. Not surprisingly, QSVA EnsVAR shows better results as a probabilistic estimator and also for more standard resolution measures. This is unsurprising, to my mind, because this comparison is not fair. As the Authors noted, the costly QSVA extension is needed to keep EnsVAR assimilation in an approx. linear error evolution regime and thus guarantee good behaviour in this long-window assimilation set-up. To compare apples with apples the Authors should directly compare results of the standard EnsVAR algorithm at the end of the window with those of the EnKF and PF. Additionally, it would also be of interest to compare results of QSVA EnsVAR with those of an EnKF whose assimilation is run on shorter assimilation windows, to guarantee linear behaviour, and then cycled.

   We have actually exactly done in Section 3, relative to strong-constraint assimilation, what the referee requests, viz., performing the comparison between the results of EnsVAR, EnKF and PF at the final times of the assimilation windows. That is now made more explicit.
   Now, concerning Section 4 of the paper, relative to weak-constraint assimilation, it is true that our comparison has been made, not at the final time of the assimilation windows, but on the last 13 days of the windows (see Fig. 12). In that sense, the comparison is not fair. A perfectly fair comparison may be included in a future work.
   The referee also mentions a 'cycled' EnKF. We do not understand what he means. To us, the EnKF, which is sequential, is cycled by construction.

2. In Sect. 4 on weak-constraint assimilation, I understand that the model error perturbations are drawn from the same error distribution whose covariance is used in the 4D-Var cost function. If this is correct, this is a significant limitation on the potential applicability of the results, as the difficulty in obtaining realistic characterizations of Q is probably the most important cause of the limited success of weak-constraint 4D-Var in realistic applications.

The referee raises a very important question, which applies equally to the EnKF and PF. But that question goes well beyond the scope of our present papers. An assimilation algorithm must first be evaluated in conditions where the errors affecting the data follow the same statistics (first- and second-order moments) that are used in the assimilation. That is what we have done. The same question arises but may be more critical concerning observation errors. We have mentioned explicitly that our twin experiments are fully 'consistent' as concerns the errors on the data.

3. In the last paragraph of Sect. 4, the Authors explain that the performance of EnsVAR, EnKF, PF in the weak-constraint case appears in terms of reliability measures (e.g., rank histograms). This could depend on localization used in the EnKF, for example. Have the Authors explored this parameter space?

   This remark is appropriate. It would be of course impossible to explore the full space of parameters for the three algorithms. Now, concerning localisation of the EnKF, and at the explicit request of referee 2 (see his comment 36 on paper 1), we have performed experiments without localisation in the setup of paper 1. The results are summarised in the new version of the latter. They show changes (some features of the assimilated fields, including the rank histograms, are improved, while others, such as the RMS errors, are degraded). These results, as interesting as they are, cannot however be studied in detail in the frame of our present work.

4. Lines 310-311: "...many possibilities exist for the reducing the cost of EnsVAR, through simple parallelization or ...". I am puzzled on how parallelization can reduce the computational cost of EnsVAR. Maybe the Authors meant clock time?

   Yes, you're right. Thanks for the remark. correction done.

5. Lines 319-320: "On the other hand, EnsVAR is largely empirical, with the consequence that, should difficulties arise, conceptual guidelines may be missing to solve these difficulties." I struggle to see what these difficulties might be. In the linear case, EnsVAR (aka EDA) is constructed so as to be a consistent statistical estimators assuming the input data errors are correctly sampled. In the nonlinear case, its behaviour will depend on the amount of nonlinearity and the ability to track the true global solution. In this respect, EnsVAR is as empirical as the EnKF.

   Well, one must always be ready to encounter unexpected difficulties. And, yes, EnKF is also empirical, and the remark we make about EnsVAR also applies to EnKF (with the additional difficulty that the latter contains many more arbitrary parameters than our EnsVAR). What we want to stress is that it is easier to interpret the results produced by a method built on a solid theoretical basis, and to correct its possible weaknesses. We have mentioned that our remark applies as well to the EnKF and PF.

6. Lines 339-340: "EnsVAR has been implemented here on a small dimension system. It has to be implemented on larger dimension, physically more realistic models.".

I suspect the Authors mean QSVA-EnsVAR in this context. Standard EnsVAR has been running at ECMWF and Météo-France for a number of years.

*Yes, our statement was not correct in the sense that EnsVAR-EDA is operationally running at ECMWF and Météo-France on large models. But, it has not been systematically evaluated as a probabilistic estimator on a physically realistic large dimensional model. That is what we consider necessary, either in the form of QSVA-EnsVAR or otherwise. Indeed, the fact that EnsVAR-EDA, although largely empirical, is run operationally with undisputably some success is one additional reason for studying its properties more deeply.*

**2  To referee 2:**

We thank M. Bocquet for his comments and suggestions. These are printed below in black, and our responses in red.

1. Abstract, line 1: You should mention here that EnsVAR is equivalent to EDA. The main issue is the confusion that it may generate, and the fact that, because you fail to refer to EDA in the abstract, you will restrict your potential readership.

   *Thanks, we have made the appropriate changes in both abstracts.*

2. Abstract, lines 3-4: If you had cycled the analysis, you would have observed that QSVA is not as mandatory, expect maybe for very long windows. So I believe you should mitigate the statement.

   *Yes, many possibilities can be considered. For instance, performing the assimilation over long enough overlapping successive windows (but that is not really different from QSVA). This is mentioned in the conclusion, but has not been used in the paper. We have modified the abstract to simply say that the problem of assimilation over long windows has been solved in our setting by using QSVA.*

3. Abstract, line 9: "without need to resort to QSVA" → "without the need for QSVA".

   *Thanks, done.*

4. line 19: "Kuramuto" → "Kuramoto", as well as in both references by Yoshiki Kuramoto et al.

   *Thanks, done.*

5. lines 24-25: "The performance of EnsVAR is compared with that of Ensemble Kalman Filter and Particle Filter in Section 3.": again, out of a specific context, this does not make much sense in the absence of cycling, proper tuning of the methods, and so on.

   *We have mentioned (ll. 305-310) that the comparison with EnKF and PF certainly cannot be considered as definitely conclusive. But it is certainly instructive,*

for instance in that it suggests that there are no major differences between the results produced by the three methods that have been compared. And we do not understand why the referee considers that 'this does not make much sense in the absence of cycling' (see our response to his specific remark 34 about paper 1). And 'proper tuning of the methods' could be an endless task.

6. line 28: "successful in nonlinear as in linear conditions.": it always depends on how long the data assimilation window is. As any other method, EnsVAR is bound to fail for very large windows.

We qualified our statement by saying that it is valid only for the conditions of our experiments. But is not clear to us why any method is bound to fail for very large windows. Failure is certainly to be expected for strong constraint assimilation implemented with an erroneous model. But why should it be in the case of weak constraint ?

7. line 33: "twice a day": please mention that this corresponds to 0.10 time units since the Lorenz model is primarily defined in those units.

We have defined the "day" in paper I as equal to 0.44 time unit in equation I.12.

8. line 35: Fine with the "I." but the notation for referring to equations is not consistent throughout the manuscript and does not follow the Nonlinear Processes in Geophysics guidelines.

Thank you, we went thoughout the manuscript and made the corrections.

9. line 59: "the the" →"the".

Thanks, done.

10. Section 2: Nice results. Similar and consistent results have been obtained, which should be briefly mentioned. Bocquet and Sakov (2013) have obtained very similar results with the iterative ensemble Kalman smoother (IEnKS) with the same window of 10 days, a time-interval of 1 day (as opposed to twice a day), an ensemble of 20 members and $\sigma = 1$: see Figure 4 of Bocquet and Sakov (2013). In particular the MDA IEnKS (S $= 1$), which is quasi-static, outperforms the SDA IEnKS which (in this reference) is not quasi-static. Other directly relevant references worth citing about quasi-static EnVar methods are Goodliff et al. (2015) and Carrassi et al. (2017).

Thanks for mentioning those references. We have included them in our conclusion. On the other hand, they do not seem to deal directly with what is the primary goal of our paper, namely the objective evaluation of algorithms for ensemble assimilation as probabilistic estimators.

11. lines 82-83: "This improvement must be due to the fact that more observations have been used.": above all this is due to the fact that the middle point is farther apart from the end of the window, so that fresher observations have a strong

information content leveraged by the unstable modes of the dynamics. This has been shown in Bocquet and Sakov (2014).

Thanks also for mentioning. We have added the reference and mentioned this pertinent remark in the paper.

12. line 105: "bayesianity" → "Bayesianity".

13. line 106: "bayesian" → "Bayesian".

Thanks, done for both.

14. lines 102-113: Part of this analysis coincides with that of H. Abarbanel and his collaborators. I believe you should at least refer to one of their paper, for instance Ye et al. (2015).

Thanks again. The reference has been added.

15. line 115: "shown on Figure 5" → "shown in Figure 5".

Thanks, done.

16. line 129: "As the errors in the ensemble means...": I believe you mean "error standard deviations of the ensemble means".

Thanks, done.

17. lines 151-153: My experience is that, on the contrary, rank histograms of a deterministic EnKF (ETKF specifically) are $\bigcap$-shape (the ensemble is overdispersed). The difference might be due to the nature of the EnKF, the fact that your EnKF run is not long enough, or simply, that your inflation is insufficient. Moreover, once again, localisation is unnecessary with an ensemble of 30 members and may be detrimental to the quality of the ensemble. Anyway, the $\bigcup$-shape that you have obtained for the EnKF is note a generality.

We take note of the fact that what we observe is not a generality, but cannot really say more at this stage. Concerning the length of the assimilation window, the top left panel of Figure 7 suggests that the EnKF is already stabilized after 3 days of assimilation, so that it seems unlikely that a longer window would significantly modify the histogram (the diagnostics which the referee comments have been performed at the end of the assimilation window).

18. line 167: b is a notation usually reserved for a possible bias in VarBC.

The notation b for bias is by no means a standard.

19. line 175, Eq. (2): Assuming H is linear, it should read H.

Thank you, done.

20. line 182: Dot missing at the end of the sentence.

Thanks, done.

21. 195: "which corresponds to a predictability time of about 10 days": interesting. Can you please develop?

   This value of 10 days results from numerical tests which were not shown in the paper. That is now briefly mentioned.

22. lines 252-255: How did you implement model noise in the EnKF and the PF? This should be described.

   As said now, the model noise has been added as random noise in all model integrations.

23. line 263: "simple :" → "simple:".

   Thanks, done.

24. lines 277-279: Yes, that is the most important added value of this couple of papers and should be emphasised in the abstract of the first manuscript.

   We think that this point has been properly, if succinctly, explained in the abstract, which we do not want to overload.

25. lines 294-296: In general, no claim can be made as to the accuracy of these methods (with the goal to estimate the truth) in the absence of cycling;

   This seems to repeat the referee's comment 5 above. See our response there.

26. lines 305-310: I already know for a fact (Bocquet and Sakov, 2013, 2014) that proper cycling would very significantly reduce the number of iterations. This should be mentioned.

   We thank again the referee for this information, which we will take into account in a possible future work.

27. lines 328-330: "is cycling necessary at all, or can one simply proceed by implementing EnsVAR over successive, possibly overlapping, windows?": This question has already a detailed answer in (Bocquet and Sakov, 2013, 2014) and subsequent references. To anticipate a question: yes, many of the conclusions obtained with the IEnKS would apply to EnsVAR. In essence: no, it is not absolutely necessary, but it would numerically help a lot to cycle the background (fewer iterations) and would yield a better accuracy.

   Thanks once more. We have added the references.

28. line 338: "(Carrassi, Bocquet, pers. com.)": this has now been published (Bocquet and Carrassi, 2017).

   Thanks, done

29. line 347: "ROT" → "RTO".

   Thanks, done

30. lines 348-349: "This defines a theoretical improvement on EnsVAR, based on an appropriate use of the Jacobian of the data operator." Liu et al. (2017) have already shown on a higher dimensional example that RTO might become inefficient (it is likely to be ultimately subject to the curse of dimensionality) as reported in their experiments and conclusions. This could be mentioned.

All right. Thanks. This is now mentioned.

---

## Referee Report (RR1)

Review of *Ensemble Variational Assimilation as a Probabilistic Estimator. Parts I and II (revision)* by M. Jardak and O. Talagrand Sumitted to Nonlin. Processes Geophys. A. Carrassi Editor.

Marc Bocquet

July 18, 2018

I thank the authors for their answers and their revision. I recommend acceptance of both manuscripts. There are a few points about which the authors might want to give it a second thought. In the following, I am not commenting on the choices made by the authors, even when I would have done otherwise. I only comment when I believe they did not get something significant. If need be, any technical correction that the authors would like to make can be made prior to uploading the final draft, under the control of the Editor.

**1 Ensemble Variational Assimilation as a Probabilistic Estimator. Part I**

- 3. Abstract, line 10-14: the emphasis is on the performance (accuracy) of the method compared to, e.g., the EnKF. I do not believe that this is wise in the absence of a proper cycling with which the EnKF could shine. I do not understand why the emphasis is not on the discussion of the Bayesian (or not Bayesian) trait of the method and the quality of the updated ensemble, which is the strong point of this study. By writing performance, we did not mean specifically numerical accuracy. We meant global performance of the algorithms under comparison, and primarily their performance as Bayesian estimators. We have modified the wording so as to avoid any misunderstanding.

  You did not really answer to this point. You now mention both Bayesian estimator and deterministic estimator. Hence, my remark is still valid. In the absence of cycling of the EnsVar, you have not shown that the EnsVar outperforms the EnKF, at least as a deterministic estimator. To be clear: the comparison that you make in the abstract is the outcome of your experiments. But these experiments do not reflect the general goal of data assimilation for the geofluids, which require cycling over a long period of time. I think this is totally acceptable for the Bayesian estimator, but not for the deterministic estimator. There is nothing wrong with your results but you should contextualise them and mention their limitation.

- 9. line 103-104: The connection between RTO and EDA as used in geophysical data assimilation has first been made, put forward and discussed in Liu et al. (2017) (and much more1). This must be mentioned here. (Incidentally this is how the authors of the present manuscript became aware of Bardsley et al. (2014).) More- over, Oliver et al. and were the first to discuss this problem in 1996 (Oliver et al., 1996), which is something that Liu et al. (2017) recalled. You must cite this reference as well. We have added all these references, and commented on their connection with the present work.

  You should mention the name RTO here too, since it is used by several authors.

- 19. line 291: It is worth mentioning that this time-step is 0.05 time unit. We have introduced the "day" as equal to 0.44 time unit in equation 12.

  My remark is still valid. Your presentation is unclear and unconventional. The reader has no way to know at this stage what this time-step is.

- line 369: "where one day is equal to 0.44 time unit...": I would rather say 0.20 time unit(?). That is actually our definition of a 'day'.

  This is not Lorenz's definition of a day for his model, and not the definition used by all of your colleagues. This is very confusing.

- lines 450-451: "Fair comparison is therefore possible only at the end of the assimilation window.": yes, but not only. A fair comparison of DA methods would also imply cycling, which is not the case of EnsVAR here. I am very fine about your using the EnKF and particle filter to compare the ensemble qualities; but not really when it comes to comparing RMSE at the end of the window. At the very least this should be briefly discussed. We do not really understand what the referee means here. What we mean is that it is only at the end of the assimilation window that the three algorithms have used the same amount of information, and that it is only at that time that comparison is fair, in terms of Bayesianity as well as RMSE. Having a form of cycling for EnsVAR within the overall assimilation window would define another algorithm, which could also be compared to what we have obtained. But we do not understand in what that would be 'fairer'.

  You have to contextualise your results. Your comparison between the EnsVar and the EnKF is established in the restricted context of your numerical experiments. It turns out that your experiments are not that general because you do not cycle EnsVar. Hence you cannot make general claim about the EnKF. A the very least, you have to recall the absence of cycling of the EnsVar. Again, the primary goal of data assimilation in meteorology/oceanography is the long-term tracking of the state of a geofluid. You have not defined an algorithm suited for this (since it cannot extend beyond a certain time horizon). I am fine with your choice. But you have to discuss this and contextualise.

- 35. line 459: "...is the one described by Evensen (2003)": which one? G. Evensen's book describes both stochastic and deterministic EnKFs. (Of course I know the answer, you just need to improve the statement.) By the way, you should, from time to time, insists on the fact that you picked up the stochastic EnKF since the deterministic is now more popular. Moreover, choosing the stochastic EnKF makes sense in this study as the EnsVAR is also stochastic. You could mention this as this would strengthen your choice for the stochastic EnKF. Yes, we have used a stochastic EnKF. This has now been added in the paper. Now, we do not see any necessary connection between the stochastic character of our two algorithms, if 'stochastic' only means that the data are perturbed at some stage in the algorithm.

  Yes, 'stochastic' only means that the data are perturbed at some stage in the algorithm. This algorithmic connection is very strong, and very well know. Try comparing with a deterministic EnKF: you will understand then!

**2 Ensemble Variational Assimilation as a Probabilistic Estimator. Part II**

- lines 24-25: "The performance of EnsVAR is compared with that of Ensemble Kalman Filter and Particle Filter in Section 3.": again, out of a specific context, this does not make much sense in the absence of cycling, proper tuning of the methods, and so on. We have mentioned (ll. 305-310) that the comparison with EnKF and PF certainly cannot be considered as definitely conclusive. But it is certainly instructive, for instance in that it suggests that there are no major differences between the results produced by the three methods that have been compared. And we do not understand why the referee considers that 'this does not make much sense in the absence of cycling' (see our response to his specific remark 34 about paper 1). And 'proper tuning of the methods' could be an endless task.

  Please see my response to the first paper authors' response. It is the sentence "and its performance is as least as good as" that causes trouble because it is not put into the rather narrow context of the experiments of the paper. No, 'proper tuning of the methods' is not an endless task (but certainly tedious), otherwise we would not be bench-marking EnKF methods. This is usually a requirement.

- line 28: "successful in nonlinear as in linear conditions.": it always depends on how long the data assimilation window is. As any other method, EnsVAR is bound to fail for very large windows. We qualified our statement by saying that it is valid only for the conditions of our experiments. But is not clear to us why any method is bound to fail for very large windows. Failure is certainly to be expected for strong constraint assimilation implemented with an erroneous model. But why should it be in the case of weak constraint ?

  As opposed to the EnKF, nobody has ever been able to set up a (single analysis) 4D-Var for chaotic models over very long windows. For the strong constraint 4D-Var, it is due to the dynamical

instability of the model. For the weak-constraint 4D-Var, this is due to the numerical cost (storage, computational). Even for stable dynamics (atmospheric chemistry for instance), it is customary to split the 4D-Var analyses over several segments of a few weeks each.

- 7. line 33: "twice a day": please mention that this corresponds to 0.10 time units since the Lorenz model is primarily defined in those units. We have defined the "day" in paper I as equal to 0.44 time unit in equation I.12.

  The day has been defined by Lorenz as 0.20 time unit. Most colleagues stick to this (not so arbitrary actually) definition. Your choice may generate a lot of confusion.

---

## Author Response (AR2)

**Answers to referees and editor : npg-2018-5 &6, 2018**

Mohamed Jardak & Olivier Talagrand

**1   To referee I:**

We thank the referee for his suggestions for future research, concerning in particular the numerical cost of EnsVAR.

**2   To referee II:**

1. The referee has spotted an inconsistency in our paper, for which we thank him. We have made the correction. Our 'day' is equal to 0.24 time unit in Equation (12) (instead of 0.2 in the paper by Lorenz). We do not think the difference is critical.

2. As requested by the referee, we have explicitly mentioned the Random-Then-Optimize (RTO ) algorithm in our Introduction (it was already mentioned in our Conclusion).

3. The other comments of the referee bear on what he considers are limitations of our work and of our conclusions. We basically agree with him, and we had already mentioned that our conclusions are limited to the conditions of our experiments. The referee mentions deterministic versions EnKF as an alternative to the stochastic version we have used. We now include the use of deterministic EnKF among the various possibilities for future works. The referee stresses that our EnsVAR is not cycled, and seems to consider that, because of the ensuing numerical cost, it could not be used in practical situations. That may the case, and cycling is already discussed in our papers, in particular in the perspective of future works. In the other hand, we do not understand some of the remarks made by the referee on this aspect of cycling. He writes for instance I think this is totally acceptable for the Bayesian estimator, but not for the deterministic estimator (where this designates our approach). We do not understand why the referee makes here a difference between the two estimators. In our logic, assimilation is intrinsically a problem in Bayesian estimation, and a deterministic estimator can only be a by-product (e.g., an expectation) of a Bayesian estimator. In any case, the fact that our EnsVAR is not cycled is stressed in two places in our Part I, and discussed again in the Conclusion of Part II. We do not think it is necessary to add more on this aspect.

**3   To the editor:**

The Editor has specifically asked us to consider two points. One is the question of the time unit we use. We think this has now been clarified. Our 'day' is equal to 0.24 time unit in Equation (12) (see point 1 in our response to Referee 2). As for the other point, the editor writes I am also concerned about the lack of a clear assertion that your experiments do not include any cycling. Well, the fact that our experiments do not include any cycling was clearly asserted in the latest version of Part I of our papers (ll. 223-225 and 387-388) and discussed again in the Conclusion of Part II (ll. 344-355). See also point 3 of our responses above to Referee 2.

**4   Reference:**

Lorenz, E. N.:, Predictability: A problem partly solved. In: Proc. Seminar on Predictability, Vol. 1. ECMWF: Reading, Berkshire, UK, pp. 1–18, 1996.